# AmbiRefer3D: 3D Visual Grounding with Referential Ambiguity

**Rongjiang Zhu** [1 2 *] **Wei Kang** [2 1 *] **Zeqi Liu** [1 2] **Junyu Chen** [1] **Shuo Yang** [2 †] **Xinxiao Wu** [1 2 †]

## Abstract

Traditional 3D visual grounding typically assumes that natural language expressions unambiguously refer to target objects in a 3D scene. However, in practical applications, human instructions are often ambiguous or insufficient, which may lead existing models to associate the query with multiple possible objects, resulting in incorrect results. In this paper, we propose a new task, 3D visual grounding with referential ambiguity, which allows for referential ambiguity in language descriptions, making it more broadly applicable to real-world scenarios. To tackle this task, we propose an interactive grounding framework that performs multi-round question-answer interactions, in which the model actively generates clarifying questions and receives human-provided answers to acquire additional object attributes, spatial relationships, and other contextual information, thereby resolving referential ambiguity and achieving accurate grounding. To support the learning of interactive grounding, we construct a large-scale dataset named AmbiRefer3D, which contains 47,085 samples with 141,255 annotations of question-answer dialogues that capture interactive disambiguation processes, covering 7,316 indoor 3D scenes. Furthermore, we establish multi-round evaluation metrics to measure both disambiguation efficiency and grounding accuracy. The code is available at *https://github.com/yearnallover/AmbiRefer3D*.

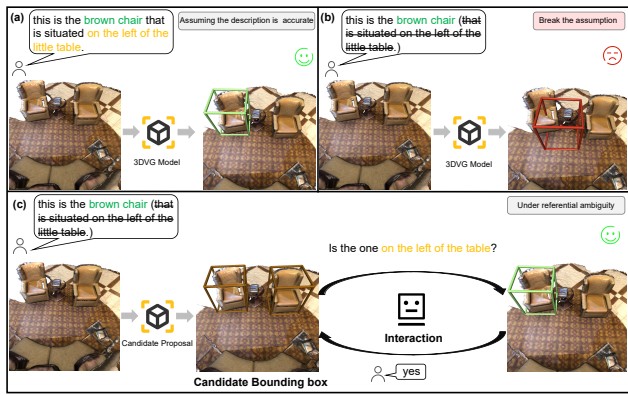

*Figure 1.* (a) Traditional 3D grounding methods assume that natural language expressions sufficiently locate specific objects. (b) Ambiguous expressions often match incorrect objects. (c) Our formulation handles referential ambiguity through a multi-round human-participated question-answer interaction.

## 1. Introduction

3D visual grounding aims to localize target objects in a 3D scene based on natural language descriptions, and serves as a fundamental component for a wide range of embodied and scene-centric tasks, including 3D scene understanding (Wang et al., 2025b), interactive manipulation (Bohg et al., 2017), and embodied interaction (Cong et al., 2021; Zhu et al., 2025).

Driven by large-scale benchmarks (Chen et al., 2020; Achlioptas et al., 2020; Zhang et al., 2023), prior works (Jain et al., 2025; Zhang et al., 2024; 2023; Chen et al., 2020) have achieved strong performance with an assumption that input referring expressions provide sufficient information to localize specific objects in the scene, as illustrated in Figure 1(a). However, this assumption frequently breaks down in real-world environments. As shown in Figure 1(b), human instructions are often ambiguous or insufficient, especially in complex scenes containing multiple objects with similar attributes or spatial relations. In such cases, the initial query may correspond to several possible objects, causing existing grounding models to produce incorrect predictions. We analyze the impact of such referential ambiguities through experiments in Appendix A.

These observations highlight the need for a more general formulation of 3D visual grounding that can explicitly handle

---

[*]Equal contribution , [†] Corresponding authors. [1]Beijing Laboratory of Intelligent Information Technology, School of Computer Science and Technology, Beijing Institute of Technology, Beijing, China [2]Guangdong Provincial Laboratory of Machine Perception and Intelligent Computing, Shenzhen MSU-BIT University, Shenzhen, China. Correspondence to: Shuo Yang <yangshuo@smbu.edu.cn>, Xinxiao Wu <wuxinxiao@bit.edu.cn>.

*Proceedings of the 43rd International Conference on Machine Learning*, Seoul, South Korea. PMLR 306, 2026. Copyright 2026 by the author(s).

referential ambiguity. To this end, we define a new task, 3D visual grounding with referential ambiguity, which allows for ambiguous language descriptions. This formulation better aligns with realistic downstream 3D applications, such as Embodied Question Answering (Majumdar et al., 2024), Vision-Language Navigation (Lin et al., 2025), and Robot Instruction Following (Kim et al., 2024), where human instructions are often inherently ambiguous.

To tackle this task, we propose an interactive 3D visual grounding framework that resolves referential ambiguity through multi-round question–answer interactions, as shown in Figure 1(c). The framework first retrieves candidate objects consistent with the referring expression, then proactively poses discriminative questions (e.g., questions about color, shape, or spatial relationships) to refine the understanding of the target, receives human answers, and iteratively updates the candidate set until the specific target object is uniquely identified. This interactive paradigm shifts 3D visual grounding from passive matching to active ambiguity resolution, making it more suitable for user-centered practical 3D applications.

To facilitate the learning of interactive grounding, we construct a large-scale dataset, AmbiRefer3D, which contains 47,085 samples and 141,255 ambiguity-aware annotations, covering 7,316 indoor scenes. Each annotation records how an ambiguous query is clarified through multi-round question–answer dialogues in a 3D environment, encompassing diverse ambiguity sources such as object attributes, spatial relationships, or their complex compositions commonly found in real-world 3D scenes. Furthermore, we introduce standardized multi-round evaluation metrics for comprehensively assessing interactive grounding performance. These metrics jointly evaluate disambiguation efficiency and grounding accuracy, reflecting the model's ability to resolve ambiguities with fewer interactions and ultimately localize the target object.

The contributions of this paper are summarized as follows:

- We propose a novel task, 3D visual grounding with referential ambiguity, which breaks the assumption of unambiguous language queries in traditional 3D visual grounding and provides a more general solution for practical applications.

- We propose an interactive grounding framework that proactively poses questions by understanding the 3D scene and utilizes human answers to progressively acquire discriminative information, achieving efficient and precise grounding.

- We build AmbiRefer3D, a large-scale dataset comprising diverse ambiguity cases across realistic and synthetic 3D scenes, and introduce standardized multi-

round evaluation metrics to support research on interactive grounding with referential ambiguity.

**Conflict of Interest Disclosure.** We declare no financial conflicts of interest related to this work.

## 2. Related Work

**Datasets of 3D Visual Grounding** 3D visual grounding aims to localize target objects in a 3D scene given natural language expressions, and advancements in datasets have significantly driven progress in this field. Early work (Chang et al., 2015; Chen et al., 2018) primarily focuses on language grounding on isolated 3D objects instead of complete scenes. To address the above limitation and support scene-level 3D visual grounding, ScanRefer (Chen et al., 2020) and ReferIt3D (Achlioptas et al., 2020) pioneer the construction of large-scale datasets based on real-world indoor scan data from ScanNet (Dai et al., 2017). Subsequent datasets enhance the practicality of 3D visual grounding in several ways, such as providing more diverse referring expressions and instruction styles to test language generalization (Achlioptas et al., 2020; Wang et al., 2025a), extending annotations beyond a single object instance to cover richer grounding targets and supervision information (Wang et al., 2026), and relaxing the one-expression-one-object setting by allowing an expression to correspond to zero, one, or multiple target instances (Zhang et al., 2023).

Despite these advancements, existing 3D grounding datasets still assume unambiguous queries, while real-world human instructions are often ambiguous or underspecified, limiting their applicability to ambiguity-aware tasks. In this paper, we propose AmbiRefer3D, a large-scale ambiguity-aware 3D grounding dataset with rich multi-round question-answer annotations across thousands of indoor scenes, and provide standardized multi-round metrics to evaluate both disambiguation efficiency and grounding precision.

**Interactive Disambiguation for Referential Grounding** Interactive disambiguation plays a crucial role when an ambiguous query may match multiple plausible objects. In 2D vision-language and conversational search tasks, multi-round interaction resolves this uncertainty by asking clarifying questions and using user-provided answers to add missing constraints, thus progressively narrowing down the intended referent (De Vries et al., 2017; Das et al., 2017; Rao & Daumé III, 2018; Aliannejadi et al., 2019). In 3D vision-language tasks, recent studies (Wei et al., 2025) incorporate disambiguation-related supervision by strengthening referring descriptions during dataset construction. In vision-language-action tasks, interactive grounding is further studied to cope with imperfect language during action execution, where interaction helps compensate for missing or noisy cues (Zhang et al., 2025).

This paper formulates the task of 3D visual grounding with referential ambiguity, and further proposes a general disambiguation framework together with a new dataset, AmbiRefer3D, providing multi-round clarification supervision and enabling systematic evaluation of ambiguity resolution in 3D environments.

## 3. Task Definition

We propose a new task, termed *3D visual grounding with referential ambiguity*, which relaxes the conventional assumption that a referring expression uniquely identifies a target object. In this task, the initial language description may correspond to multiple objects in a 3D scene.

Formally, a 3D scene is represented as a point cloud $P \in \mathbb{R}^{N \times D}$, where $N$ denotes the number of points and $D$ denotes spatial coordinates and per-point features (e.g., color, normals). Let $\mathcal{O}$ denote the set of all objects in the scene. Given a referring expression $T$, the goal is to localize a target object $o^* \in \mathcal{O}$. Unlike conventional 3D visual grounding, the expression $T$ is *ambiguous* and is compatible with a subset of objects $\mathcal{C} \subseteq \mathcal{O}$, where

$$\mathcal{C} = \{o \in \mathcal{O} \mid o \text{ satisfies } T \text{ in scene } P\}. \quad (1)$$

To resolve the ambiguity, additional information $\mathcal{E}$ is provided to further constrain the target object. The task objective is to identify the target by reasoning over the scene $P$, the initial expression $T$, and the additional constraints $\mathcal{E}$, formulated as

$$o^* = \arg\max_{o_i \in \mathcal{C}} \mathcal{F}(o_i \mid P, T, \mathcal{E}), \quad (2)$$

where $\mathcal{F}(\cdot)$ denotes a grounding function.

This task formulation emphasizes disambiguative grounding, where multiple plausible objects must be distinguished using additional constraints.

## 4. Method

To tackle the task of 3D visual grounding with referential ambiguity, we propose an interactive 3D visual grounding framework, which consists of two stages: candidate generation and interactive disambiguation. In the stage of candidate generation, we retrieve an exhaustive set of candidate objects $\mathcal{C}$ that are semantically consistent with the ambiguous query $T$. In the stage of interactive disambiguation, we design multi-round question-answer interactions. At each round, we first perform constraint selection to analyze the current candidate set and identify the most discriminative object attributes or spatial relationships to pose a question for clarification. Then, upon receiving human answers, we perform candidate filtering to incorporate additional infor-

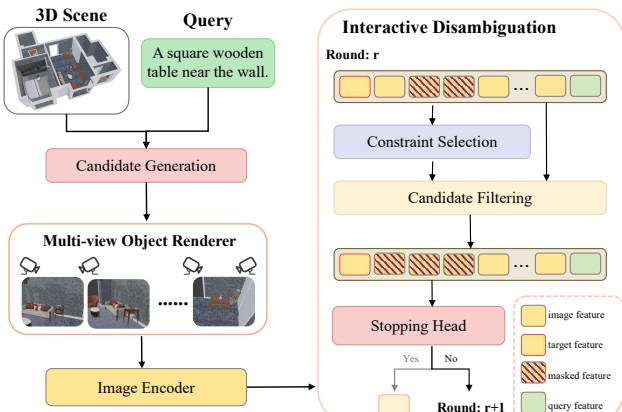

*Figure 2.* An overview of our framework.

mation to dynamically prune the candidate set $\mathcal{C}$. The interaction is stopped until the target object is identified or the maximum round is reached. An overview of our framework is illustrated in Figure 2.

### 4.1. Candidate Generation

Given a 3D scene represented as a point cloud $P$ and a referring expression $T$, the candidate generation stage aims to retrieve a set of potential target objects, denoted by $\mathcal{C} = \{o_i\}_{i=1}^N$. We use existing multi-object 3D grounding models for candidate generation, which first extract a comprehensive set of object proposals and then select the proposals relevant to the referring expression as the final candidate objects. Specifically, the proposals whose semantic relevance scores exceed a predefined confidence threshold $\tau$ are considered relevant to $T$.

### 4.2. Interactive Disambiguation

Since the initial referring expression $T$ may correspond to multiple plausible objects, we introduce an *interactive disambiguation* stage to resolve referential ambiguity through multi-round question-answer interactions. This stage iteratively introduces discriminative constraints, progressively narrowing down the set of candidate objects until the target object is uniquely identified. Specifically, at each interaction round $r$, the interactive disambiguation performs three steps: (1) visual representation extraction, (2) constraint selection, and (3) candidate filtering. The interaction terminates when the *Stopping Head* predicts a confidence exceeding a threshold $\beta$, or when the maximum number $R$ of rounds is reached.

**Visual Representation Extraction**. To obtain visual representations, we render $M$ multi-view images from different viewpoints for each candidate object $o_i \in \mathcal{C}$, and encode them using a ViT-B/32 image encoder (Dosovitskiy et al., 2021). We then average-pool the features from different

viewpoints to form an object-level visual representation $\mathbf{v}_i \in \mathbb{R}^{d_v}$, and stack the feature vectors of all candidate objects to obtain the feature matrix $\mathbf{V} \in \mathbb{R}^{|\mathcal{C}| \times d_v}$.

**Constraint Selection.** The goal of constraint selection is to identify the most informative clarification information that can maximally reduce ambiguity among the current candidate objects. To this end, we tokenize and encode the original referring expression $T$ using the text encoder from the 3D grounding model of the candidate generation stage, generating a sentence-level embedding $\mathbf{t} \in \mathbb{R}^{d_t}$. Then, we compute a global state representation $\mathbf{s}^{(r)}$ by pooling visual features over the current candidate objects $\mathcal{C}^{(r)}$ together with the query embedding $\mathbf{t}$. Finally, a clarification constraint $g^{(r)}$ is selected that is expected to maximally reduce referential ambiguity.

Concretely, we first formulate the constraint selection as choosing a clarification constraint from a predefined discrete constraint space $\mathcal{G}_{\text{cons}}$, where each constraint is semantically grounded in the 3D scene and parameterized as structured tuples (type, slot, value) regarding object attributes and spatial relationships. Then we design a *constraint selector* that operates on the current set of candidate objects and outputs a single set-level representation for selecting the most discriminative constraint under the remaining ambiguity. This selector first aggregates information from current candidate objects, and then utilizes current context and explicit variability cues to evaluate the extent to which each constraint can differentiate the candidate objects.

To capture the current contextual information of the candidate objects, we compute a set-level feature $\mathbf{v}_{\text{valid}}^{(r)}$ by mean pooling over the features of current candidate objects, which summarizes the approximate center of the candidate set and we compute $\mathbf{v}^{(r)} = \{\mathbf{v}_i\}_{o_i \in \mathcal{C}^{(r)}}$ denoting the visual features of current candidates at round $r$. Beyond this global contextual information, we characterize target-centered variability by constructing $\boldsymbol{\mu}_{\text{diff}}^{(r)}$ and $\boldsymbol{\delta}_{\text{max}}^{(r)}$ based on the target object representation $\mathbf{v}_{\text{target}}$, where $\boldsymbol{\mu}_{\text{diff}}^{(r)}$ measures the average discriminative margin between the target object and the current candidate objects, and $\boldsymbol{\delta}_{\text{max}}^{(r)}$ captures the most distinct candidate object relative to the target object. These signals collectively guide the selector toward clarifications, thereby maximally reducing the ambiguity of the target object.

Finally, the selector combines the global state representation $\mathbf{s}^{(r)}$, the query embedding $\mathbf{t}$, current visual features $\mathbf{v}^{(r)}$, $\mathbf{v}_{\text{target}}$, the average difference $\boldsymbol{\mu}_{\text{diff}}^{(r)}$, and the maximum difference $\boldsymbol{\delta}_{\text{max}}^{(r)}$, to predict the most discriminative constraint $\mathbf{c}^{(r)}$, formulated by

$$\mathbf{c}^{(r)} = \text{Selector}\left(\mathbf{s}^{(r)}, \mathbf{t}, \mathbf{v}^{(r)}, \mathbf{v}_{\text{target}}, \boldsymbol{\mu}_{\text{diff}}^{(r)}, \boldsymbol{\delta}_{\text{max}}^{(r)}\right). \quad (3)$$

**Candidate Filtering.** Given a selected constraint $c^{(r)}$, the goal of candidate filtering is to update the current set of candidate objects by eliminating objects that are incompatible with $c^{(r)}$.

Concretely, we implement the candidate filtering as a neural *mask head* that performs constraint-conditioned candidate object evaluation. This mask head takes the visual feature of each candidate object, the query embedding, and the semantic representation of the selected constraint as input. To enable flexible reasoning over different types of clarifications, the mask head adopts a cross-attention mechanism, where the candidate object features attend to the joint representation of the query and the constraint embedding. This allows the model to evaluate the compatibility of candidate objects with the selected constraint in a context-aware manner. Formally, for each candidate object $o_i \in \mathcal{C}^{(r)}$, the mask head predicts its corresponding compatibility score:

$$m_i^{(r)} = \text{MaskHead}(\mathbf{v}_i, \mathbf{t}, g^{(r)}), \quad (4)$$

where $\mathbf{v}_i$ denotes the visual feature of $o_i$, $\mathbf{t}$ is the query embedding, and $g^{(r)}$ is the semantic embedding of the selected constraint. The compatibility scores are mapped to probabilities via a sigmoid function $\sigma(\cdot)$, and the candidate objects that violate the constraint are pruned using a fixed threshold of probability $\alpha$. That is, the candidate objects whose probabilities exceed $\alpha$ are reserved as the updated set, given by

$$\mathcal{C}^{(r+1)} = \{ o_i \in \mathcal{C}^{(r)} \mid \sigma(m_i^{(r)}) \geq \alpha \}. \quad (5)$$

By iteratively applying constraint-conditioned filtering, the candidate set is progressively reduced, enabling multi-round ambiguity resolution until a confident decision can be made.

### 4.3. Training Objective

The training loss of our framework is formulated by

$$\mathcal{L} = \mathcal{L}_{\text{G}} + \lambda_{\text{sel}}\mathcal{L}_{\text{sel}} + \lambda_{\text{cand}}\mathcal{L}_{\text{cand}} + \lambda_{\text{stop}}\mathcal{L}_{\text{stop}}, \quad (6)$$

where $\mathcal{L}_G$ is a grounding loss used in the stage of candidate generation, $\mathcal{L}_{\text{sel}}$ is a constraint selection loss that supervises the model to predict the ground-truth clarification constraint in each interaction round, $\mathcal{L}_{\text{cand}}$ is a candidate ranking loss that encourages the target object to receive higher scores than other candidate objects under the selected constraint, and $\mathcal{L}_{\text{stop}}$ is a stopping loss that predicts whether the disambiguation process should terminate in the current round. Detailed formulations of all loss terms are provided in Appendix B.

## 5. AmbiRefer3D Dataset

To facilitate the research on 3D visual grounding with referential ambiguity, we construct AmbiRefer3D, a large-scale dataset based on ScanNet (Dai et al., 2017) and 3D-FRONT (Fu et al., 2021). Unlike traditional 3D visual

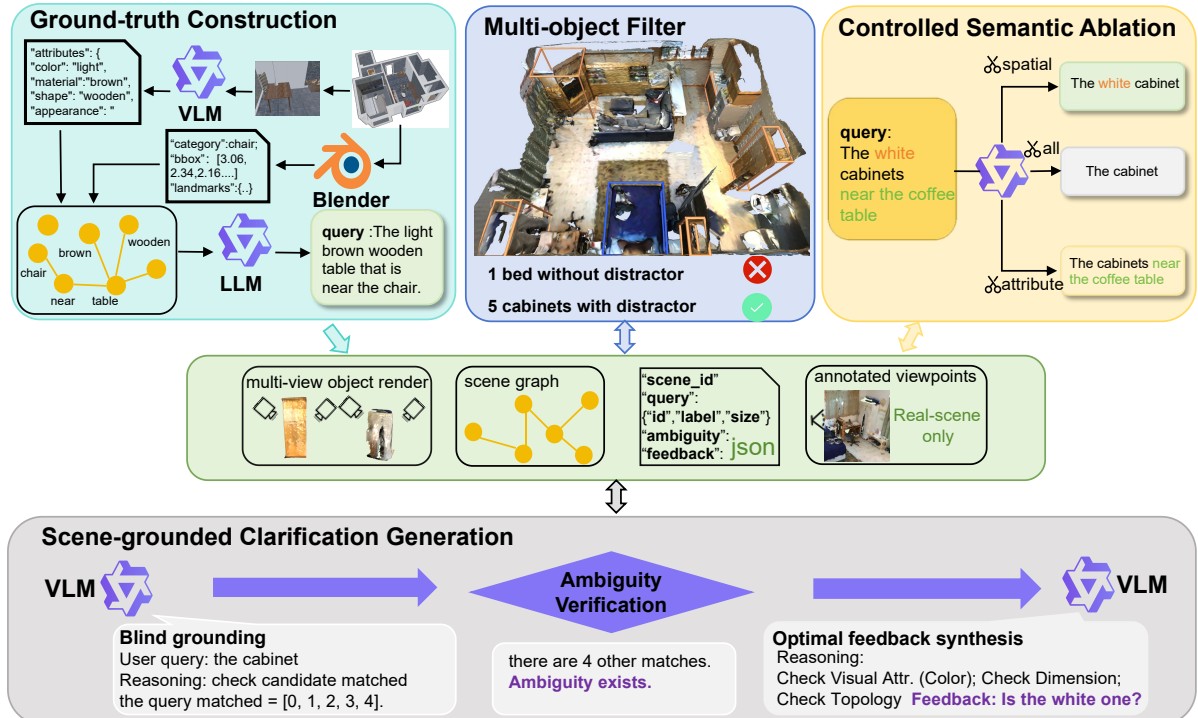

*Figure 3.* The pipeline for AmbiRefer3D data generation.

grounding datasets, AmbiRefer3D focuses on scenarios where the initial referring expression is ambiguous and corresponds to multiple candidate objects. To support active disambiguation, each ambiguous query is paired with a sequence of question–answer dialogues that simulate a scene-grounded disambiguation process, ultimately leading to the unique identification of the target object.

Specifically, AmbiRefer3D covers three types of referential ambiguity, including (1) *Attribute*, where the referential expression lacks discriminative visual attributes; (2) *Spatial Relation*, where the referential expression lacks the necessary spatial relationships to distinguish the target; and (3) *Attribute and Spatial Relation*, where both attribute and spatial information are missing. For each case, we construct a clarification process that leverages scene context to generate informative disambiguation questions, ultimately transforming the ambiguous expression into a unique reference. Several examples of different types of reference ambiguity are shown in Figure 4.

### 5.1. Data Generation

AmbiRefer3D is constructed through a unified data generation pipeline, combining both real-world and synthetic 3D scenes, and introducing scene-type-specific components only when necessary. The overall pipeline is illustrated in Figure 3.

**Ground-truth Construction.** For real-world scenes, we use manually written referring expressions from the ScanRefer dataset as ground-truth queries. For synthetic scenes, we use the 3D-FRONT dataset, where referring expressions are not provided. Therefore, we construct ground-truth queries directly from the scene specifications.

Specifically, we obtain object-level attributes by rendering individual objects and querying a vision-language model with the rendered images. We further use precise object attributes and positional information gained from the 3D renderer to construct a scene graph that contains object categories, 3D bounding boxes, and inter-object relationships, following the structured representation of SceneVerse (Jia et al., 2024). Based on this scene graph, we generate unique, scene-grounded referring expressions using a constraint-based LLM generation process.

**Multi-object Filter.** To ensure that ambiguity can arise, we design a multi-object filter that keeps only queries whose target object co-occurs with at least one other object of the same class in the scene.

**Controlled Semantic Ablation.** To construct ambiguous referring expressions, we systematically remove discriminative information from ground-truth queries with LLM assistance. This process aims to introduce referential ambiguity while preserving semantic validity with respect to the underlying scene, resulting in three types: *Attribute*, *Spatial Relation*, and *Attribute and Spatial Relation*. These

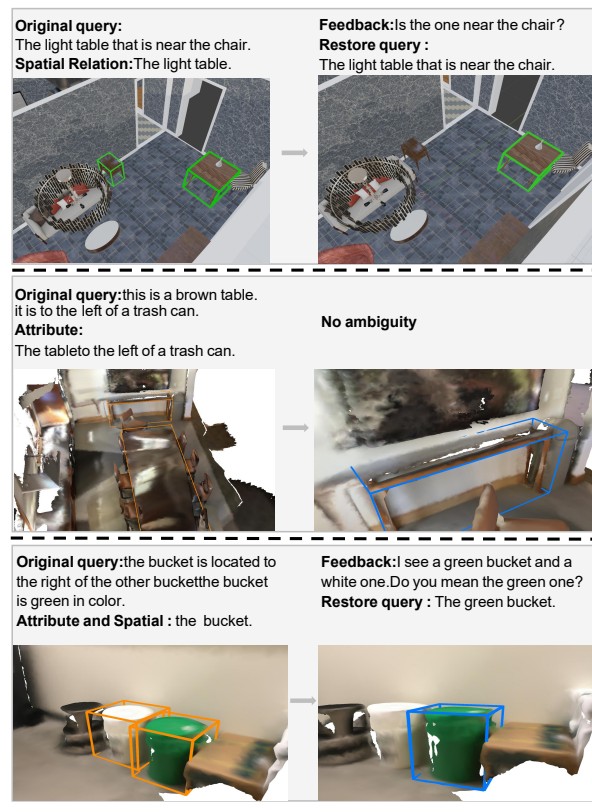

Figure 4. Examples of different types of referential ambiguity on AmbiRefer3D.

Table 1. Dataset statistics of AmbiRefer3D.

| | |
|---|---|
| Scenes | 7316 |
| Initial ambiguous queries | 141255 |
| Clarification steps | 113904 |
| Avg. clarification steps per object | 2.42 |
| Objects | 47085 |
| Avg. objects per scene | 6.44 |
| Avg. candidate objects per query | 2.89 |
| Attribute | 29953 |
| Spatial Relationship | 39059 |
| Attribute and Spatial Relationship | 44882 |

Table 2. Comparison with existing 3D visual grounding datasets.

| Dataset | Scenes | Descriptions | Ambiguity | Clarification Feedback |
|---|---|---|---|---|
| ScanRefer | 800 | 51583 | ✗ | ✗ |
| Nr3D | 707 | 83572 | ✗ | ✗ |
| Sr3D | 707 | 41503 | ✗ | ✗ |
| ScanScribe | 1185 | 278 | ✗ | ✗ |
| ScanReason | 1456 | 12929 | ✗ | ✗ |
| ViGiL3D | 35 | 350 | ✗ | ✗ |
| Multi3DRefer | 800 | 61926 | ✗ | ✗ |
| AmbiRefer3D | 7316 | 141255 | ✓ | ✓ |

*feedback synthesis* using a hierarchical discriminative strategy aimed at maximizing information gain. Specifically, the intrinsic visual attributes (e.g., color, shape) are prioritized whenever possible, while the extrinsic spatial relationships derived from geometric cues are introduced only when visual attributes are insufficient. This design encourages the construction of efficient clarification constraints rather than exhaustively enumerating all missing information.

For synthetic scenes, candidate objects corresponding to an ambiguous query are first explicitly identified by traversing the scene graph and enforcing geometric and semantic consistency constraints. Due to the deterministic nature of the synthetic scene representation, this step generates an accurate and exhaustive set of candidate objects whenever ambiguity exists. If multiple valid candidate objects exist, we directly apply the same *feedback synthesis* strategy used for real-world scenes to construct clarification constraints. More details about the entire data generation process can be found in Appendix E.

### 5.2. Dataset Statistics

AmbiRefer3D contains 7,316 indoor 3D scenes and a total of 141,255 ambiguity-aware referring annotations. Each sample corresponds to a referring expression in a 3D scene. Expressions that match multiple objects are treated as ambiguous and form a candidate set for grounding, while expressions that match a single object are regarded as unambiguous. There are three types of referential ambiguity: *Attribute*, *Spatial Relation*, and *Attribute and Spatial Relation*, reflecting missing visual attributes, missing spatial relations,

yield a spectrum of underspecified queries that correspond to multiple possible objects in the scene.

**Scene-grounded Clarification Generation.** To provide a reliable process for uniquely identifying the target object from an ambiguous referring expression, we first identify a set of possible objects compatible with the ambiguous query. Then, based on these objects and their surrounding environments, we associate each ambiguous query with a series of scene-grounded clarification constraints. These constraints introduce additional discriminative information grounded in the 3D scene, enabling the query to uniquely localize the target object.

For real-world scenes, we adopt a generate-verify-synthesize process to construct valid and informative clarification constraints. Given an ambiguous query, the generator is provided with a structured multimodal context containing (1) object-centric visual renderings, (2) semantic relationships derived from the scene graph, and (3) geometric priors computed from the underlying 3D layout combined with the annotation perspective. To eliminate hallucinations, the generator first performs *blind verification* by attempting to ground the query. A query is considered ambiguous only when the retrieved multiple candidate objects contain the target object. Following verification, the generator performs

*Table 3.* Main results on real-world scenes of AmbiRefer3D with different ambiguity types. $Ours\dagger$ indicates an upper-bound setting where ground-truth candidate sets from the candidate generation stage are provided during training.

| Ambiguity | Method | TS | Acc | S@1 | S@3 | S@6 | AvgR |
|---|---|---|---|---|---|---|---|
| Attribute | M3DRef-CLIP | - | 34.28 | - | - | - | - |
| | M3DRef-CLIP + Ours | 60.7 | 39.44 | 24.64 | 30.24 | 39.44 | **1.28** |
| | D-LISA | - | 14.68 | - | - | - | - |
| | D-LISA + Ours | 68.7 | 39.02 | 26.94 | 32.22 | 39.02 | 1.52 |
| | Ours† | - | **45.02** | **37.98** | **42.67** | **45.02** | 1.46 |
| Spatial Relation | M3DRef-CLIP | - | 34.24 | - | - | - | - |
| | M3DRef-CLIP + Ours | 66.7 | 37.52 | 21.77 | 29.31 | 37.52 | 2.41 |
| | D-LISA | - | 9.55 | - | - | - | - |
| | D-LISA + Ours | 72.2 | 36.88 | 26.78 | 32.78 | 36.88 | **2.10** |
| | Ours† | - | **41.53** | **24.00** | **37.30** | **41.53** | 2.31 |
| Attribute + Spatial Relation | M3DRef-CLIP | - | 33.24 | - | - | - | - |
| | M3DRef-CLIP + Ours | 75.7 | 37.21 | 16.98 | 27.03 | 37.21 | 3.56 |
| | D-LISA | - | 5.21 | - | - | - | - |
| | D-LISA + Ours | 77.9 | 34.02 | **22.63** | 27.82 | 34.02 | 3.82 |
| | Ours† | - | **43.78** | 18.31 | **32.56** | **43.78** | 3.27 |
| Mixture | M3DRef-CLIP | - | 36.15 | - | - | - | - |
| | M3DRef-CLIP + Ours | 65.0 | 40.03 | 27.85 | 36.20 | 40.03 | 3.51 |
| | D-LISA | - | 9.81 | - | - | - | - |
| | D-LISA + Ours | 72.1 | 41.57 | 26.07 | 36.15 | 41.57 | 3.39 |
| | Ours† | - | **49.51** | **30.41** | **40.35** | **49.51** | 3.12 |

and missing both visual attributes and spatial relations, respectively. Statistics on query types, candidate set size, and clarification steps are summarized in Table 1. We compare AmbiRefer3D with existing 3D visual grounding datasets in Table 2, highlighting its unique support for ambiguity-aware grounding. Additional analyses and fine-grained statistics are provided in Appendix E.

### 5.3. Evaluation Metrics

To comprehensively evaluate the performance of interactive grounding on AmbiRefer3D, we establish evaluation metrics covering two dimensions: candidate object retrieval quality and disambiguation efficiency.

**Candidate Retrieval.** We report Target Survival (TS) to evaluate the quality of the initially retrieved candidate objects. TS measures the proportion of samples in which the ground-truth target object is successfully included in the set of candidate objects. High TS is a prerequisite for successful disambiguation.

**Disambiguation Efficiency.** To specifically measure the efficiency and accuracy of ambiguity resolution, we propose three multi-round metrics. First, we use Success@$K$(S@$K$) to evaluate efficiency, which measures the percentage of samples where the target object is correctly identified within at most $K$ clarification rounds. Second, to quantify the interaction cost, we compute Average Rounds (AvgRounds) as the mean number of rounds required to successfully identify the target object. Third, we report Accuracy (Acc) as the final performance metric, representing the proportion of samples where the final prediction result correctly localizes the target object after interactions.

## 6. Experiments

### 6.1. Implementation Details

In the candidate generation stage of the proposed framework, M3DRef-CLIP (Zhang et al., 2023) and D-LISA (Zhang et al., 2024) are used as the backbone and fine-tuned on the AmbiRefer3D training set. The confidence threshold for candidate retrieval is set to $\tau = 0.1$. In the interactive disambiguation stage, the number of rendered multi-view images for each candidate object is set to $M = 3$. At each round of disambiguation, the fixed sigmoid threshold for candidate filtering is set to $\alpha = 0.4$. The stop threshold is set to $\beta = 0.5$. The maximum number of interaction rounds is set to $R = 6$. During training, we adopt the AdamW optimizer with a learning rate of $3 \times 10^{-6}$ and a weight decay of $1 \times 10^{-4}$, and the batch size is set to $32$.

### 6.2. Main Results

To evaluate the effectiveness and generality of the proposed interactive grounding framework, we conduct comparison experiments on AmbiRefer3D under four ambiguity types: *Attribute*, *Spatial Relation*, *Attribute + Spatial Relation*, and *Mixture*, where the first three correspond to individual ambiguity sources and *Mixture* denotes joint training on all ambiguity types. Note that M3DRef-CLIP and D-LISA are directly applied to ambiguous queries without interaction for comparison, and they are also the backbones of candidate generation in our framework.

Table 3 shows the results on real-world scenes, and the results on synthetic scenes can be found in Appendix C. We observe that directly applying existing 3D visual grounding models results in substantially degraded performance across all ambiguity types, with particularly severe drops under *Spatial Relation* and *Mixture*. This observation confirms that traditional grounding models, which assume unambiguous language, struggle to handle realistic, ambiguous scenarios. In contrast, incorporating the same backbones into our interactive framework consistently improves grounding accuracy and Success@K across all ambiguity types. Notably, *M3DRef-CLIP+Ours* and *D-LISA+Ours* achieve significant performance gains under the more challenging *Attribute + Spatial Relation* and *Mixture*, where ambiguity is more pronounced. Meanwhile, the average number of interaction rounds remains low, indicating that our framework effectively resolves ambiguity efficiently.

The performance gap between *Ours* and *Ours*† reflects the impact of candidate generation quality on the overall system, and also demonstrates that the interactive disambiguation effectively narrows down the candidate set when reliable candidate objects are available.

Overall, these results demonstrate that our framework generalizes well to different grounding backbones, maintains

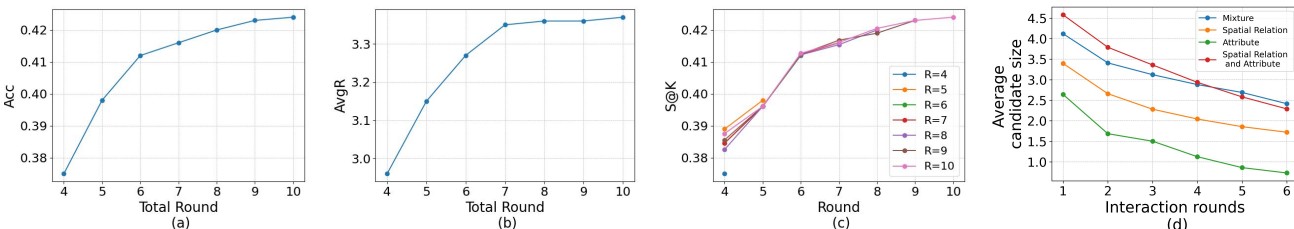

*Figure 5.* Interactive disambiguation dynamics across rounds. (a) Final accuracy under different maximum interaction rounds; (b) Average number of rounds required for successful grounding; (c) Cumulative success rate (Success@$K$) as the interaction progresses; (d) Average candidate set size per interaction round, we set $R = 6$. All results are obtained using *D-LISA+Ours*, trained on all ambiguity types.

robustness across various ambiguity types, and provides an effective solution to referential ambiguity in realistic 3D scenes.

### 6.3. Interactive Disambiguation Analysis

To further evaluate the effectiveness of the interactive disambiguation, we analyze how grounding performance and candidate set dynamics evolve across rounds, as illustrated in Figure 5. Specifically, we conduct this analysis using the *D-LISA+Ours* method, and vary the maximum number of interaction rounds $R$ from 4 to 10, to investigate whether ambiguity can be effectively resolved within a limited number of interaction steps, and how additional interaction rounds contribute to performance gains.

As shown in Figure 5 (a) and Figure 5 (b), the overall grounding accuracy consistently improves as the number of allowed interaction rounds increases. However, the trend of improvement gradually saturates, indicating that most ambiguous cases can be resolved within a small number of rounds. This demonstrates that our framework is capable of completing disambiguation efficiently, without relying on excessively long interactions.

Figure 5 (c) reveals that allowing additional interaction rounds still benefits a subset of hard samples, as reflected by the increasing Success@K scores. Figure 5 (d) shows that the average size of the candidate set decreases rapidly in the early rounds and continues to shrink steadily thereafter. This result confirms that the proposed constraint selection and candidate filtering effectively eliminate candidate objects early in the interaction process, resulting in more efficient ambiguity resolution.

Overall, these results demonstrate that our interactive disambiguation framework achieves a good balance between grounding accuracy and interaction efficiency. To further analyze the effectiveness of interactive disambiguation, we provide additional experiments and detailed analyses in Appendix C.

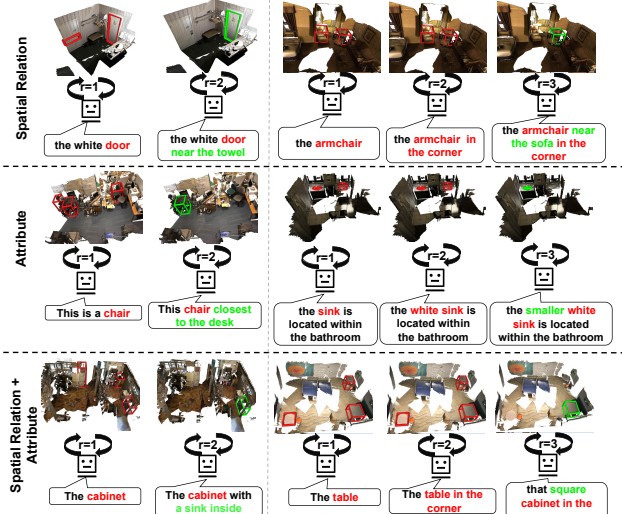

*Figure 6.* Qualitative evaluation results of our framework in three ambiguity types, red text and bounding boxes indicate ambiguous referring expressions, and their corresponding candidate objects, green text and bounding boxes denote discriminative information and the final resolved target.

### 6.4. Ablation Study

**Loss Components.** To verify the contribution of each training objective in our interactive disambiguation framework, we conduct an ablation study on the loss components, as shown in Table 4.

We observe that the three loss terms play complementary roles in the interactive disambiguation process. Removing $L_{stop}$ leads to the highest Acc, as the model is no longer constrained to stop early and can continue interaction until the target is correctly grounded. However, this comes at the cost of a higher AvgR, indicating reduced interaction efficiency. This reflects a trade-off between accuracy and efficiency, where $L_{stop}$ primarily regulates when to terminate the interaction. In contrast, removing either $L_{cand}$ or $L_{sel}$ results in clear performance degradation. Without $L_{cand}$, the model fails to effectively filter distractors, preventing progressive reduction of the candidate set. Without $L_{sel}$, the model cannot select informative constraints, leading to sub-

Table 4. Ablation study on loss components.

| $L_{sel}$ | $L_{cand}$ | $L_{stop}$ | Acc | S@1 | S@3 | S@6 | AvgR |
|---|---|---|---|---|---|---|---|
| ✓ | ✓ | ✗ | **42.36** | 25.89 | **36.44** | **42.36** | 3.74 |
| ✓ | ✗ | ✓ | 38.13 | **26.13** | 33.08 | 38.13 | 4.12 |
| ✗ | ✓ | ✓ | 24.47 | 7.40 | 8.06 | 24.47 | 5.16 |
| ✓ | ✓ | ✓ | 41.57 | 26.07 | 36.15 | 41.57 | **3.39** |

Table 5. Ablation study on $v_{target}$ selection strategies. We compare random selection, selecting the highest-scoring candidate, and mean aggregation over candidates.

| Selection of $v_{target}$ | Acc | S@1 | S@3 | S@6 | AvgR |
|---|---|---|---|---|---|
| Random Selection | 28.92 | 10.41 | 16.88 | 28.92 | 4.16 |
| Mean Aggregation Selection | 21.09 | 5.71 | 11.14 | 21.09 | 4.34 |
| Highest-scoring(**Ours**) | **41.57** | **26.07** | **36.15** | **41.57** | **3.39** |

Table 6. Ablation study on constraint selector strategies, including random selection, heuristic selection, and an all-at-once strategy that applies all constraints in a single round.

| Constraint Selector Strategy | Acc | S@1 | S@3 | S@6 | AvgR |
|---|---|---|---|---|---|
| Random Selection | 23.31 | 8.53 | 9.16 | 23.31 | 4.88 |
| Heuristic Selection | 33.76 | 24.48 | 30.21 | 33.76 | 2.76 |
| Top-$K$ Selection | 37.88 | **30.42** | 35.69 | 37.88 | **1.89** |
| **Ours** | **41.57** | 26.07 | **36.15** | **41.57** | 3.39 |

optimal and inefficient queries. These results highlight that effective interaction requires jointly optimizing constraint selection, candidate filtering, and stopping decisions.

**Selection Strategy.** To evaluate the impact of different approximations, we conduct ablation studies with alternative selection strategies for approximating in Table 5 , including: Random, Mean Aggregation, and Highest-scoring(Ours).

Results show that the highest-scoring candidate achieves the best performance. This confirms that our scoring mechanism effectively identifies the most informative target representation, leading to improved grounding performance.

**Constraint Selector Strategy.** We further analyze the impact of different constraint selection strategies on interactive disambiguation. We compare our adaptive multi-round selector with three alternative strategies: Random Selection, which randomly chooses a constraint at each round; Heuristic Selection, which greedily selects the constraint that reduces the candidate set the most; and Top-$K$ Selection, which applies the top-$K$ predicted constraints simultaneously in one round, with $K = 3$ in our experiments.

As shown in Table 6, Top-$K$ selection can accelerate early-stage disambiguation, achieving relatively strong performance at S@1 (30.42). However, its gains at later rounds remain limited. This is because our dataset is inherently designed with multiple complementary constraints, where each instance typically involves three or more constraints that must be jointly satisfied. When top-ranked constraints are applied together in one round, the model can benefit from stronger initial pruning without accumulated interaction error. However, as more constraints are jointly introduced, their effects may potentially interfere with each other, limiting further improvement at S@3 and S@6.

## 6.5. Qualitative Results

To better understand how our framework resolves referential ambiguity, Figure 6 illustrates representative qualitative examples across different ambiguity types. Regardless of whether the ambiguity stems from missing attributes, spatial relations, or a combination of both, each interaction round consistently identifies a clarification constraint that is highly discriminative with respect to the current candidate objects. These examples demonstrate that the proposed constraint selection effectively utilizes scene-grounded visual and spatial cues to progressively prune inconsistent candidate objects, thereby achieving accurate and efficient grounding. These results indicate that our framework follows a systematic process, consistently focusing on discriminative constraints that reduce ambiguity across diverse real-world conditions.

## 7. Conclusion

In this paper, we introduce a new task, 3D visual grounding with referential ambiguity, which allows for referential ambiguity in language descriptions, making it more broadly applicable to real-world scenarios. We propose an interactive grounding framework that performs multi-round question-answer interactions, resolving referential ambiguity and achieving efficient and precise grounding performance. To support research on interactive grounding with referential ambiguity, we construct a large-scale dataset, AmbiRefer3D, which covers diverse sources of referential ambiguity in indoor 3D scenes and provides supervision for multi-round disambiguation, and introduce standardized multi-round evaluation metrics. Extensive experiments demonstrate both the necessity of our proposed task and the effectiveness of our framework. We believe our efforts will facilitate future research on grounding in realistic 3D environments, contributing to improving 3D reasoning tasks.

## Acknowledgments

This work was supported by the Shenzhen Science and Technology Program under Grant No.JCYJ20241202130548062, Guangdong Provincial Key Area Project of General Universities under Grant No.2025ZDZX3049 and Grant No.2024ZDZX1017.

## Impact Statement

This paper introduces a new task and framework for 3D visual grounding under referential ambiguity, with the goal of advancing research on more realistic and robust grounding tasks. The primary contribution of this work is methodological and empirical, focusing on task formulation, framework construction, dataset construction, and evaluation metrics for interactive ambiguity resolution.

We do not anticipate any immediate negative societal impacts arising from this work. The proposed dataset and framework are designed for research purposes and do not involve sensitive personal data or deployment in high-stakes decision-making scenarios. Potential applications of this research, such as embodied agents, human-robot interaction, and assistive systems, may benefit from improved robustness to ambiguous instructions. As with many advances in machine learning, downstream applications should be developed and deployed responsibly, with appropriate consideration of safety, transparency, and user intent.

We believe this work contributes positively to the development of more reliable and human-aligned AI systems, and does not raise significant ethical concerns beyond those commonly associated with machine learning research.

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

# A. Task Validity and Dataset Effectiveness Analysis

To empirically validate the necessity of explicitly modeling referential ambiguity in 3D visual grounding, we conduct a controlled study on real-world 3D scenes using representative single-target 3D visual grounding methods. This experiment is designed to support the motivation illustrated in Figure 1, which highlights the discrepancy between conventional grounding benchmarks and realistic, ambiguous scenes.

## A.1. Experimental Setup

We select five representative single-target 3D visual grounding methods, denoted as UniVLG (Jain et al., 2025), TSP3D (Guo et al., 2025),ConcreteNet (Unal et al., 2024), M3DRef-CLIP (Zhang et al., 2023), ScanRefer (Chen et al., 2020), and evaluate them under three different query settings on real-world 3D scenes:

- **(a) Ground-Truth Query.** The original human-annotated referring expressions from ScanRefer (Chen et al., 2020), which are assumed to uniquely identify the target object.

- **(b) Ambiguous Query.** Ambiguous referring expressions constructed by removing attribute, spatlial and all of them from the original queries. These expressions are designed to reflect realistic instructions that commonly occur in natural human communication, where a single expression may correspond to multiple possible objects in the scene.

- **(c) Disambiguated Query.** Clarified referring expressions obtained by augmenting ambiguous queries with minimal additional information, as we described in the main paper, such that the target object becomes uniquely identifiable again. These queries simulate the outcome of successful human clarification.

All methods are evaluated using standard grounding accuracy, measuring whether the predicted object matches the ground-truth target.

## A.2. Results and Analysis

Table 7 reports the grounding accuracy of each method under the three query settings. Across all evaluated methods, performance under the ambiguous query setting (b) drops substantially compared to the ground-truth setting (a), demonstrating that existing single-target grounding models are highly sensitive to referential ambiguity. In contrast, when disambiguated queries (c) are provided, model performance largely recovers to a level comparable to that of ground-truth queries.

*Table 7.* Grounding accuracy (@0.5) of single-target 3D visual grounding methods under different query settings.

| Method | Ground-Truth | Ambiguous | Disambiguated |
|---|---|---|---|
| UniVLG (Jain et al., 2025) | 44.3 | 33.8 | 43.0 |
| TSP3D (Guo et al., 2025) | 40.9 | 32.8 | 37.0 |
| ConcreteNet (Unal et al., 2024) | 38.8 | 30.7 | 34.7 |
| M3DRef-CLIP (Zhang et al., 2023) | 36.0 | 30.2 | 34.5 |
| ScanRefer (Chen et al., 2020) | 19.0 | 16.0 | 18.2 |

These results indicate that the observed performance degradation is not due to inherent limitations of the grounding models themselves, but rather stems from the mismatch between their single-target formulation and the ambiguous nature of realistic language instructions. Furthermore, the strong performance recovery under disambiguated queries validates the effectiveness and realism of our query construction process.

Overall, this study provides empirical evidence that referential ambiguity poses a fundamental challenge to conventional 3D visual grounding formulations, and motivates the need for datasets and methods that explicitly model ambiguity and its resolution, as proposed in this work.

## B. Loss Function Details

### B.1. Candidate Generation Loss

The candidate generation models in the first stage are trained following the original objectives defined in their respective 3D visual grounding frameworks. In this work, we adopt two representative multi-object grounding models, M3DRef-CLIP (Zhang et al., 2023) and D-LISA (Zhang et al., 2024), both of which build upon a shared detection backbone and optimize a combination of detection, reference, and cross-modal alignment objectives.

For Multi3DRefer, the overall training objective is defined as:

$$\mathcal{L}_G = \mathcal{L}_{\text{det}} + \mathcal{L}_{\text{ref}} + \mathcal{L}_c, \tag{7}$$

where $\mathcal{L}_{\text{det}}$ denotes the detection loss inherited from PointGroup (Jiang et al., 2020), $\mathcal{L}_{\text{ref}}$ is the reference loss supervising object-description matching, and $\mathcal{L}_c$ is a symmetric contrastive loss that aligns sentence features with object-level visual features.

Specifically, the detection loss $\mathcal{L}_{\text{det}}$ consists of four components: (i) a cross-entropy loss for per-point semantic segmentation, (ii) an $\ell_1$ loss for regressing offset vectors towards object centers, (iii) a cosine similarity loss for constraining offset directions, and (iv) a binary cross-entropy loss for proposal objectness prediction. The reference loss $\mathcal{L}_{\text{ref}}$ supervises the association between detected objects and referring expressions, while the contrastive loss $\mathcal{L}_c$ encourages robust multi-modal alignment by pulling together visual and textual features from the same scene-instruction pair and pushing apart mismatched pairs.

D-LISA follows a similar training paradigm and employs the same detection loss $\mathcal{L}_{\text{det}}$ based on the PointGroup backbone. For reference supervision, D-LISA adopts a binary cross-entropy loss over detected object proposals for multi-object grounding, where the Hungarian algorithm (Kuhn, 1955) is used to establish optimal matches between predicted boxes and ground-truth objects based on pairwise IoU. A detected object is considered correctly grounded if it is matched to a ground-truth box with IoU exceeding a threshold $\tau_{\text{train}}$. For single-object grounding cases, a standard cross-entropy loss is used. In addition, D-LISA applies a symmetric contrastive loss $\mathcal{L}_{\text{ctr}}$ between object features and word features to enhance cross-modal representation learning. The total loss of D-LISA is formulated as a weighted sum of all loss terms:

$$\mathcal{L}_G = \lambda_{\text{det}}\mathcal{L}_{\text{det}} + \lambda_{\text{ref}}\mathcal{L}_{\text{ref}} + \lambda_{\text{ctr}}\mathcal{L}_{\text{ctr}} + \lambda_{\text{dyn}}\mathcal{L}_{\text{dyn}}, \tag{8}$$

where $\lambda_i$ denotes the weighting factor for each loss term.

### B.2. Interactive Disambiguation Loss

The second-stage disambiguation model is trained with single-round supervision. The overall objective consists of three loss terms corresponding to constraint selection, candidate discrimination, and stopping prediction.

$$\mathcal{L} = \lambda_{\text{sel}}\mathcal{L}_{\text{sel}} + \lambda_{\text{cand}}\mathcal{L}_{\text{cand}} + \lambda_{\text{stop}}\mathcal{L}_{\text{stop}}. \tag{9}$$

In all experiments, we set $\lambda_{\text{sel}} = 1.0$, $\lambda_{\text{cand}} = 1.0$, and $\lambda_{\text{stop}} = 0.2$.

**Constraint Selection Loss.** The constraint selection loss $\mathcal{L}_{\text{sel}}$ is formulated as a multi-class cross-entropy loss over the constraint inventory:

$$\mathcal{L}_{\text{sel}} = -\log \frac{\exp(s_{c^*})}{\sum_{c \in \mathcal{K}} \exp(s_c)}, \tag{10}$$

where $s_c$ denotes the predicted logit for constraint $c$, $c^*$ is the ground-truth clarification constraint, and $\mathcal{K}$ is the constraint inventory.

**Candidate Discrimination Loss.** The candidate discrimination loss $\mathcal{L}_{\text{cand}}$ encourages the model to assign higher confidence to the target object than to other candidates. We adopt a standard multi-class cross-entropy loss over the candidate set:

$$\mathcal{L}_{\text{cand}} = -\log \frac{\exp(s_{o^*})}{\sum_{o \in \mathcal{C}} \exp(s_o)}, \tag{11}$$

where $s_o$ denotes the predicted confidence score for candidate object $o$, and $o^*$ is the ground-truth target.

*Table 8.* Main results on synthetic scenes of AmbiRefer3D under different ambiguity settings.

| Data source | Method | TS | Acc | S@1 | S@3 | S@6 | AvgR |
|---|---|---|---|---|---|---|---|
| Attribute | M3DRef-CLIP (Zhang et al., 2023) | - | 2.52 | - | - | - | - |
| | M3DRef-CLIP + Ours | 16.9 | 10.31 | 6.70 | 8.76 | 10.31 | **2.38** |
| | D-LISA (Zhang et al., 2024) | - | 4.96 | - | - | - | - |
| | D-LISA + Ours | 31.6 | 17.20 | 11.18 | 14.62 | 17.20 | 2.12 |
| | Ours† | - | **27.80** | **22.01** | **25.93** | **27.80** | 1.88 |
| Spatial Relation | M3DRef-CLIP (Zhang et al., 2023) | - | 3.57 | - | - | - | - |
| | M3DRef-CLIP + Ours | 22.5 | 10.52 | 6.84 | 8.94 | 10.52 | 6.42 |
| | D-LISA (Zhang et al., 2024) | - | 8.99 | - | - | - | - |
| | D-LISA + Ours | 32.9 | 12.71 | 8.26 | 10.80 | 12.71 | **3.18** |
| | Ours† | - | **21.17** | **13.44** | **17.52** | **21.17** | 3.18 |
| Attribute + Spatial Relation | M3DRef-CLIP (Zhang et al., 2023) | - | 3.29 | - | - | - | - |
| | M3DRef-CLIP + Ours | 22.7 | 10.32 | 6.71 | 8.77 | 10.32 | 4.08 |
| | D-LISA (Zhang et al., 2024) | - | 6.97 | - | - | - | - |
| | D-LISA + Ours | 36.2 | 12.13 | **7.88** | 10.31 | 12.13 | 3.82 |
| | Ours† | - | **25.68** | 17.71 | **21.07** | **25.68** | **3.66** |
| Mixture | M3DRef-CLIP (Zhang et al., 2023) | - | 4.38 | - | - | - | - |
| | M3DRef-CLIP + Ours | 25.0 | 12.45 | 9.39 | 11.17 | 12.45 | 3.03 |
| | D-LISA (Zhang et al., 2024) | - | 11.74 | - | - | - | - |
| | D-LISA + Ours | 33.3 | 18.42 | 11.37 | 15.01 | 18.42 | 3.27 |
| | Ours† | - | **32.47** | **23.51** | **27.61** | **32.47** | **4.06** |

**Stopping Loss.** The stopping loss $\mathcal{L}_{\text{stop}}$ is implemented as a binary cross-entropy loss:

$$\mathcal{L}_{\text{stop}} = -\big[y \log \sigma(s_{\text{stop}}) + (1-y) \log(1 - \sigma(s_{\text{stop}}))\big], \tag{12}$$

where $s_{\text{stop}}$ is the predicted stopping logit and $y \in \{0, 1\}$ indicates whether the disambiguation process should terminate.

# C. Additional Experiment Analysis

**Experiments on Synthetic Scenes.** We further conduct experiments on the synthetic scenes of AmbiRefer3D to analyze the behavior of our framework in a fully controlled setting. Table 8 reports the main results on synthetic scenes. Since existing 3D visual grounding methods are trained on real-world scans and do not have pretrained weights of synthetic environments, a direct comparison with prior methods is not feasible. Therefore, all results on synthetic scenes are obtained by training and evaluating models within our framework and dataset. These results demonstrate that the proposed interactive disambiguation framework generalizes well to synthetic 3D scenes and remains effective under diverse ambiguity conditions.

*Table 9.* Candidate Generation results under different ambiguity settings on real-world and synthetic scenes.

| Setting | Method | Real-world Scenes | | | Synthetic Scenes | | |
|---|---|---|---|---|---|---|---|
| | | Prec. | Rec.@0.5 | F1@0.5 | Prec. | Rec.@0.5 | F1@0.5 |
| Attribute | M3DRef-CLIP | 50.5 | 35.4 | 32.0 | 13.9 | 13.6 | 13.1 |
| | D-LISA | 56.2 | 39.3 | 42.8 | 17.8 | 23.4 | 17.8 |
| Spatial Relation | M3DRef-CLIP | 55.9 | 34.7 | 40.0 | 13.4 | 15.4 | 13.9 |
| | D-LISA | 59.3 | 36.8 | 41.7 | 19.7 | 25.0 | 19.0 |
| Attribute + Spatial Relation | M3DRef-CLIP | 62.0 | 33.7 | 41.9 | 19.4 | 16.9 | 14.2 |
| | D-LISA | 62.9 | 56.2 | 57.0 | 19.7 | 26.0 | 19.1 |
| Mixture | M3DRef-CLIP | 55.0 | 34.0 | 39.2 | 14.4 | 19.0 | 16.4 |
| | D-LISA | 58.9 | 48.6 | 50.0 | 20.0 | 26.0 | 18.6 |

**Analysis of Candidate Generation.** To further investigate the role of the first-stage candidate generation, we analyze its performance on both real-world and synthetic scenes. Table 9 summarizes the candidate proposal results under different settings. We report Recall@0.5, Precision@0.5, and F1@0.5 to measure the quality of the generated candidate sets. The

results show that the candidate generation stage achieves high recall across both real and synthetic scenes, ensuring that the ground-truth target is preserved for subsequent disambiguation. At the same time, the moderate precision highlights the presence of plausible distractors, which motivates the necessity of the second-stage interactive disambiguation.

**Comparison with MLLM-based Methods.**    We build two MLLM-based baselines using Inst3D-LMM (Yu et al., 2025) and Video-3D LLM (Zheng et al., 2025). At each round, the MLLM asks a clarifying question, which is used to query Qwen3-VL-32B to answer strictly based on the target object. This interaction is repeated until the MLLM determines that the query is no longer ambiguous or a maximum of 6 rounds is reached, after which the MLLM outputs the final bounding box. We evaluate under the most challenging Mixture ambiguity setting, and the results are shown in Table 10. Our method achieves the best performance across all metrics. This demonstrates the effectiveness of our framework for resolving ambiguous 3D grounding.

*Table 10.* Comparison with MLLM-based baselines on real-world scenes under the mixture ambiguity setting.

| Method | Acc | S@1 | S@3 | S@6 | AvgR |
|---|---|---|---|---|---|
| Inst3D-LMM | 26.80 | 10.55 | 18.7 | 26.8 | 4.85 |
| Video-3D LLM | 22.35 | 8.10 | 15.42 | 22.35 | 5.24 |
| **Ours** | **41.57** | **26.07** | **36.15** | **41.57** | **3.39** |

**Inference-time Latency Analysis.**    We analyze the inference-time overhead introduced by multi-view rendering and multi-round disambiguation. Table 11 reports the latency breakdown compared with standard one-shot 3D visual grounding. Importantly, the overhead is dominated by multi-view rendering, while the per-round computation remains lightweight. The interaction itself is therefore efficient, and the increased cost mainly comes from additional rounds rather than heavy model inference. Moreover, in practical settings where multi-view images are already available (e.g., robotics or embodied agents), the rendering cost can be largely eliminated, significantly reducing overall latency.

*Table 11.* Inference-time latency comparison with standard 3DVG. Latency is measured per query-object grounding instance.

| Method | Views | AvgR | Rendering | Stage1 | Stage2 | ALL Latency | Relative Cost |
|---|---|---|---|---|---|---|---|
| M3DRef-CLIP | - | 1 | - | - | - | 82 ms | 1.00× |
| M3DRef-CLIP+Ours | 3 | 3.42 | 3102 ms | 82 ms | 122 ms | 3306 ms | 40.31× |

# D. Human Evaluation

To ensure the high fidelity of the generated dataset and the logical consistency of the disambiguation process, we conducted a systematic human-in-the-loop verification. This stage focuses on two critical aspects of our pipeline: the accuracy of the ambiguous candidate identification and the appropriateness of the interactive strategy.

### D.1. Verification Interface and Methodology

We developed a specialized interactive platform to facilitate the audit process (see Figure 7). The interface consists of two primary modules:

- **3D Spatial Visualization Viewport:** This module renders the 3D scene mesh with overlaid bounding boxes. It visualizes the **Ground Truth Set** (the target and valid distractors) alongside the model's **Predicted Set**, enabling verifiers to instantly check if the system correctly localized all objects causing the ambiguity.

- **Semantic Reasoning Panel:** This panel exposes the model's internal inference logic, displaying the initial ambiguous query, the generated disambiguation question, and a fine-grained reasoning trace. This allows experts to judge whether the system's decision to ask (or not ask) is derived from a valid reasoning process.

During verification, expert annotators cross-reference the visual evidence with the textual reasoning to ensure the dataset reflects human-aligned spatial understanding.

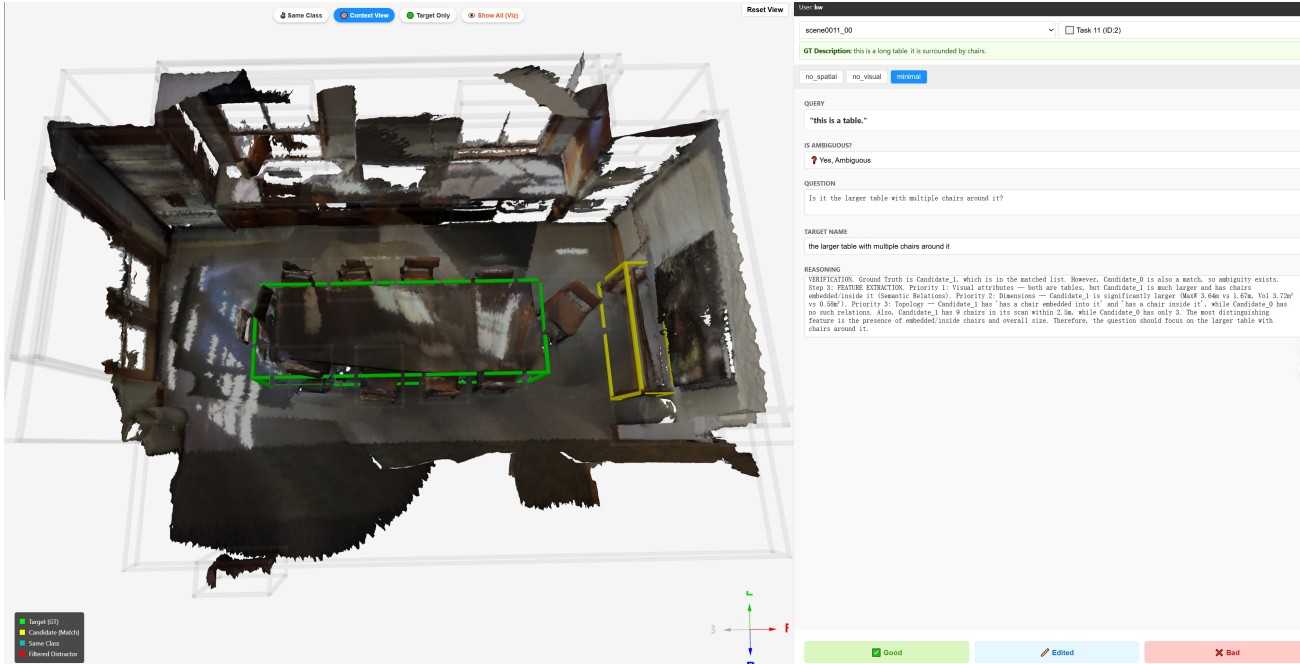

*Figure 7.* Overview of the Human Verification Interface. The left panel shows the 3D visualization of the target and distractors. The right panel displays the reasoning trace and the generated disambiguation question.

### D.2. Evaluation Metrics

The human verification is governed by two core indicators:

1. **Referential Exhaustiveness (RE):** This metric evaluates the precision of the candidate identification. It is defined as an **Exact Set Match**: the system must identify *all* potential distractors that satisfy the ambiguous query without including unrelated objects. A score of 1 is awarded only if the predicted candidate set is identical to the ground truth set; otherwise, it is 0.

2. **Disambiguation Efficacy (DE):** This metric evaluates the *appropriateness* and *quality* of the system's response. It accounts for two scenarios:

   - **Scenario A (No Ambiguity):** If the query uniquely identifies a target, the system must *not* generate a question. Success is defined as providing a direct grounding result (Null Question).
   - **Scenario B (Ambiguity Exists):** If ambiguity exists, the system must generate a question that is: (1) **Discriminative** (uniquely excludes distractors based on reasoning), and (2) **Cognitively Efficient** (uses the most salient features).

   The DE score is 1 only if the system behaves correctly according to the scenario (Correct Silence or Optimal Question).

### D.3. Results and Analysis

To quantitatively assess the quality of our dataset, we employed a random sampling strategy, selecting approximately 5% of the generated scenes for manual verification. This subset covers a diverse range of indoor environments and object configurations. Expert annotators graded each instance based on the metrics defined above.

The evaluation results are summarized in Table 12. We report the accuracy for **Referential Exhaustiveness (RE)** and **Disambiguation Efficacy (DE)**.

*Table 12.* **Human Verification Results.** Verification on a random sample ($N = 7062$). **RE**: Referential Exhaustiveness (Candidate Set Recall); **DE**: Disambiguation Efficacy (Correctness of the decision to ask and the question quality).

| Ambiguity Type | Count | RE (%) | DE (%) |
|---|---|---|---|
| Spatial Relation | 2354 | 93.5 | 88.6 |
| Attribute | 2354 | 90.9 | 93.2 |
| Attribute + Spatial Relation | 2354 | 99.6 | 84.1 |
| **Overall** | **7062** | **94.7** | **88.6** |

**Analysis.** As shown in Table 12, our pipeline achieves high fidelity across both metrics (Overall RE: 94.7%, DE: 88.6%). For **DE**, the results highlight the model's adaptability. In the *Attribute* setting, the model achieves the highest DE (93.2%), indicating it successfully leverages spatial relations to formulate discriminative questions when visual cues are absent. Conversely, the *Attribute + Spatial Relation* setting shows a slightly lower DE (84.1%). This is attributed to the higher complexity of determining the optimal feedback in the *Attribute + Spatial Relation* setting. Without initial attribute cues, the model must autonomously evaluate and rank a wider array of potential distinguishing features, ranging from visual appearance to spatial topology, to formulate the most effective question, which presents a greater reasoning challenge. Crucially, the DE metric also confirms that in non-ambiguous instances (where the description remains unique despite attribute removal), the system correctly suppresses unnecessary feedback, ensuring natural interaction.

# E. Details of AmbiRefer3D

## E.1. Details of Data Collection

AmbiRefer3D is constructed by transforming unambiguous ground-truth referring expressions into ambiguity-aware annotations with scene-grounded clarification supervision. This process is driven by a combination of Qwen3-VL-32B (Bai et al., 2025) for scene-conditioned perception and Qwen3-14B (Yang et al., 2025) for controlled linguistic rewriting and clarification generation. For transparency and reproducibility, we provide representative prompts and intermediate JSON formats used at each stage of data construction.

**Scene Filtering and Ambiguity Precondition.** For real-world scenes from ScanRefer, we first apply an object frequency filter to retain only scenes in which at least one object category appears more than five times. This filtering step ensures that each retained scene satisfies the necessary precondition for referential ambiguity, namely the coexistence of multiple possible objects of the same category. Statistics of object categories meeting this criterion are summarized in Table 13 for ScanRefer and Table 14 for 3D-FRONT, respectively.

*Table 13.* Statistics of object categories with more than 5 instances in a scene in ScanRefer. The total number of instances is 7,163.

| Category | Count | Category | Count | Category | Count |
|---|---|---|---|---|---|
| Chair | 5040 | Towel | 100 | Shelf | 38 |
| Pillow | 510 | Picture | 97 | Keyboard | 37 |
| Monitor | 362 | Window | 76 | Stool | 29 |
| Table | 166 | Armchair | 75 | Bench | 25 |
| Desk | 157 | Basket | 69 | Box | 22 |
| Cabinet | 119 | Container | 55 | Bookshelf | 11 |
| Door | 115 | Seat | 55 | Bottle | 5 |

**Ground-Truth Query Construction.** For each retained target object, we start from a canonical ground-truth query that uniquely localizes the target within the scene. In synthetic scenes, such queries are constructed directly from structured scene specifications, while in real-world scenes they are derived from original ScanRefer annotations. Representative prompts used for ground-truth query construction are illustrated in Figure 8 and Figure 9.

**Ambiguity Induction.** Starting from ground-truth queries, we systematically induce ambiguity using the controlled semantic ablation strategy described in the main paper. Specifically, we generate three types of ambiguous queries—*Attribute*, *Spatial Relation*, and *Attribute + Spatial Relation*—by selectively removing discriminative visual attributes and/or spatial

*Table 14.* Statistics of object categories with more than 5 instances in a scene in 3D-Front. The total number of instances is 245,186.

| Category | Count | Category | Count | Category | Count |
|---|---|---|---|---|---|
| cabinet | 144350 | lighting | 1543 | slabside | 112 |
| baseboard | 48342 | door | 584 | appliance | 83 |
| pocket | 15502 | column | 407 | storage unit | 79 |
| others | 13516 | sofa | 259 | wardrobe | 30 |
| cornice | 4687 | accessory | 184 | lightband | 29 |
| table | 3139 | bed | 144 | stool | 25 |
| back | 3012 | bookcase / jewelry armoire | 119 | beam | 15 |
| front | 2791 | hole | 119 | nightstand | 11 |
| chair | 2354 | plants | 114 | | |

relations. Intermediate JSON examples capturing the transformation from ground-truth queries to ambiguous queries are shown in Figure 10.

**Scene-Grounded Clarification Generation.** For each ambiguous query that matches multiple possible objects, we further generate scene-grounded clarification signals following the generate-verify-synthesize procedure. This stage ensures that ambiguity arises from the scene configuration itself and that each ambiguous query is accompanied by valid clarification constraints capable of uniquely identifying the target. Representative prompts and JSON outputs for clarification generation are visualized in Figure 11.

### E.2. Additional Dataset Statistics

We provide additional fine-grained statistics to characterize the scale and composition of AmbiRefer3D across both data sources.

**Semantic Label Alignment.** Since 3D-FRONT and ScanRefer use different semantic label taxonomies, we perform semantic label alignment before using 3D-FRONT scenes for training and evaluation. This step is necessary to ensure that objects from synthetic and real-world scenes are represented within a unified semantic space.

Specifically, 3D-FRONT provides both super-category labels and fine-grained category-level labels. We therefore adopt a two-step mapping strategy to align 3D-FRONT object labels with the NYU40 semantic taxonomy used by ScanNet/ScanRefer. First, we perform a coarse mapping based on the super-category of each object, which provides a stable initial correspondence to the NYU40 label space. Second, we refine this mapping using the object-level category label: if the fine-grained category corresponds to a valid NYU40 class, we remap the object to that class; otherwise, we retain the super-category mapping. This strategy preserves fine-grained semantic information when available while avoiding unreliable mappings for categories that do not have a clear NYU40 counterpart.

After this alignment, all 3D-FRONT instances are projected into the same NYU40 semantic space as ScanNet/ScanRefer. Categories that are absent in 3D-FRONT simply do not appear in the synthetic subset, rather than introducing a separate label system. This ensures semantic consistency across real and synthetic scenes and makes the training and evaluation protocols comparable across datasets.

**Global Statistics.** Table 15 and Table 16 summarize the overall dataset statistics for synthetic scenes (3D-FRONT) and real-world scenes (ScanRefer), respectively. These statistics include the number of scenes, ambiguous queries, clarification steps, objects, and candidate set sizes, reflecting both the scale of the dataset and the degree of ambiguity present in each source.

**Ambiguity Type Distribution.** Figure 12 and Figure 13 report the distribution of ambiguity types across the two sources. While all three ambiguity types are present in both datasets, synthetic scenes exhibit a more balanced distribution, whereas real-world scenes contain a higher proportion of spatially ambiguous queries due to denser object layouts.

To further analyze the linguistic properties of ambiguous referring expressions, we conduct a fine-grained analysis of omitted information in AmbiRefer3D. Specifically, we examine the most frequently omitted visual attributes and spatial relation

terms in *Attribute* and *Spatial Relation* queries. Figure 14 visualizes the top-$K$ omitted terms for each ambiguity type and data source. This analysis confirms that ambiguity in AmbiRefer3D arises from systematic and linguistically grounded omissions rather than from artificial or degenerate query construction.

### E.3. Qualitative Visualization

Figure 15 and Figure 16 present additional qualitative examples from real-world and synthetic scenes, respectively. In each example, the target object is highlighted in green, while other objects of the same category are shown for reference. These visualizations illustrate how ambiguous referring expressions correspond to multiple possible objects and how scene-grounded clarification cues enable progressive disambiguation.

### E.4. Failure Case Analysis

In rare instances, compromised scanning quality leads to inaccurate geometric estimation. As shown in Figure 17, this results in erroneous attribute-based disambiguation, whereas spatial relations would have remained valid.

- **User Query:** "This bookshelf is stacked."
- **Model Output (Error):** "Is the small one?" (Caused by scan artifacts)
- **Expected (Right):** "Is the one on the *left*?"

## Intrinsic Attributes Acquisition:

"You describe an object rendered on a plain background.\n"

"Return ONE short English noun phrase in the exact format:\n"

"'a <color> <material> <shape> <category>'\n"

"Rules:\n"

f"- <color> MUST be one of:
["white","black","red","green","blue","yellow","brown","gray","orange","purple", "pink",
'cyan',"magenta","beige","maroon","navy","olive","teal","lime","indigo"]\n"

f"- <material> MUST be one of:
["wooden","metal","plastic","glass","fabric","leather","ceramic",
"stone" ,"paper","rubber","concrete","marble"]\n"

f"- <shape> MUST be one of:  [ "round", "rectangular", "square", "cylindrical",
"spherical","flat", "curved", "cone-shaped", "oval", "triangular", "hexagonal", "irregular", "elliptical"]
\n"

"- <category> MUST exactly equal the given category hint (lowercase).\n"

"- Choose the dominant color/material/shape.\n"

"- Return ONLY the phrase, lowercase, no period.\n"

"Example: 'a black metal cylindrical lamp', 'a white ceramic round vase'.\n"

## Ground Truth Construction:

CATEGORY: {cat}

ROOM: {room}

LANDMARK: {lm_phrase}

EVIDENCE (may be partial):

- color: {color}

- size: {size}

- material: {material}

- shape: {shape}

- appearance_phrase: {appearance}

CANDIDATES:

{cands_text}

*Figure 8.* Prompt of Ground-truth Query Construction.

# Ground Truth Construction:

Pick one and rewrite (optional). Return JSON only.

You pick the best referring expression candidate for a 3D indoor object.

You are given:

- category, room, landmark phrase (optional)

- object attributes as evidence (color/size/material/shape/appearance)

- a list of candidate expressions

Rules:

- You MUST choose ONE candidate from the list (by index).

- You MAY slightly rewrite that chosen candidate to improve fluency,

  but you MUST NOT add any new information not supported by the evidence fields.

- If evidence is missing/unknown, do NOT mention it.

- Output JSON only.

Output format:

{

  "chosen_index": int,

  "rewrite": string

}

# Ground Truth Polishing:

ORIGINAL EXPRESSION:

"{expr}"

  "You are given a referring expression for a 3D object in an indoor scene.\n"

  "Rewrite it into ONE fluent, natural English sentence that preserves ALL original information.\n"

  "Do NOT add or remove any details.\n"

  "Do NOT introduce new reference objects.\n"

  "Do NOT introduce 'another <same category>' phrases (e.g., 'another bed') unless it already exists.\n"

  "Return plain text only."

*Figure 9.* Prompt of Ground-truth Query Construction.

**You are a specialized data generator tasked with constructing an "Ambiguous User Query" dataset.**
Your goal is to generate three distinct "degraded" variants of a given ground truth object description by selectively removing information layers. You must distinguish between **visual attributes** (color, material, shape) and **spatial information** (location, relationships).

**The Ambiguous Variants are defined as follows:**
- **No Spatial:** Retain the object class and visual attributes, but **REMOVE** all spatial context (e.g., "on the table", "next to the chair").
- **No Visual:** Retain the object class and spatial context, but **REMOVE** all visual descriptors (e.g., "red", "wooden", "tall").
- **Minimal:REMOVE BOTH** visual and spatial information. Keep only the object class and a determiner.

**Example:**
- **Input:** "The small blue ceramic vase sitting on the mantelpiece."
- **Output Variants:**
    - no_spatial: "The small blue ceramic vase."
    - no_visual: "The vase sitting on the mantelpiece."
    - minimal: "The vase."

**Respond strictly in the following JSON format:**
```
{
 "parsed_components": {
  "class": "vase",
  "visual_attributes": ["small", "blue", "ceramic"],
  "spatial_relations": ["on the mantelpiece"]
 },
 "ambiguous_variants": [
  {
   "type": "no_spatial",
   "query": "The small blue ceramic vase",
   "removed_info": ["spatial"]
  },...
 ]
}
```

*Figure 10.* Prompt of Controlled Semantic Ablation.

**You are an expert AI assistant with advanced spatial awareness and visual reasoning capabilities.**
You have a "God View" of a 3D scene. Your task is to help a user identify a specific target object. The user has provided a query that might be ambiguous (applies to multiple objects).

**YOUR GOAL:** Act as a **"Blind Listener"** to find all objects that match the user's description, then determine if the query is unique or ambiguous.

**INPUT DATA EXPLAINED:** You will receive a list of Candidate Objects with the following metadata:

**1.Visual Evidence:** Images of the object.

**2.Semantic Relations:** High-confidence relationships (e.g., "on the table", "inside the cabinet").

**3.Geometric Scan:** A raw list of neighbors with 3D vector data.

   ◦**Vector Interpretation:** Use aligned_vector [x,y,z] for horizontal direction (Left/Right/Front/Back) relative to the camera view.

   ◦**Gravity Interpretation:** Use global_dz for vertical relations (Above/Below).

**REASONING PIPELINE (Strict Step-by-Step):**

**STEP 1: BLIND GROUNDING (The "Turing Test")**

•Deconstruct the user query (e.g., "white", "cabinet", "right of table").

•Look at **ALL** Candidates. List *every* candidate that strictly matches the description based on Visuals and Spatial context.

•**Do not** use the "Hidden Ground Truth" yet.

**STEP 2: VERIFICATION & AMBIGUITY CHECK**

•Reveal the **Hidden Ground Truth (GT)** provided at the end of the prompt.

•**Check:** Is the GT in your matched list from Step 1?

•**Check:** Are there OTHER candidates in your matched list?

   ◦If **YES** (Multiple matches including GT) → **Ambiguity Exists**. Go to Step 3.

   ◦If **NO** (Only GT matches) → **No Ambiguity**. Return question: null.

**STEP 3: FEATURE EXTRACTION (If Ambiguous)**

•Compare the GT against the other matched candidates.

•Find the *most distinguishing feature* using this strict hierarchy:

   ◦**Visual Attributes:** (Color, Shape, Material).

   ◦**Dimensions:** (Significantly Taller/Wider/Larger).

   ◦**Precise Topology:** (e.g., "Candidate_0 is inside the rack, Candidate_1 is on top").

   ◦**Relative Landmarks:** (Proximity to unique objects).

•Generate a natural language clarification question based on this feature.

**CONSTRAINTS:**

**1.No Metadata Leakage:** Do not output raw vector numbers (e.g., "vector x>0.5") in the question.

**2.Natural Language:** Use "it" or "the one".

**3.No Repetition:** Do not ask about attributes already mentioned in the user query.

**OUTPUT FORMAT:** Respond strictly in JSON format:

JSON

"reasoning""Step 1: Both Cand_0 and Cand_1 match 'white chair'. Step 2: Ambiguous. Step 3: Visuals are identical, but Cand_0 is 'next to the bed' (Topology).""matched_candidate_indices"01"question""Is it the one next to the bed?""restored_query""the white chair next to the bed"

*Figure 11.* Prompt of Scene-grounded Clarification Generation.

Table 15. Dataset statistics in 3D-Front.

| | |
|---|---|
| Scenes | 6755 |
| Initial ambiguous queries | 99912 |
| Clarification steps (total) | 84050 |
| Avg. clarification steps per object | 2.34 |
| Objects (total) | 33304 |
| Avg. objects per scene | 4.93 |
| Avg. candidate objects per query | 2.91 |
| Attribute | 24078 |
| Spatial Relation | 28471 |
| Attribute + Spatial Relation | 31491 |

Table 16. Dataset statistics in ScanRefer.

| | |
|---|---|
| Scenes | 561 |
| Initial ambiguous queries | 41343 |
| Clarification steps (total) | 29854 |
| Avg. clarification steps per object | 2.17 |
| Objects (total) | 13781 |
| Avg. objects per scene | 24.57 |
| Avg. candidate objects per query | 2.84 |
| Attribute | 5875 |
| Spatial Relation | 10588 |
| Attribute + Spatial Relation | 13391 |

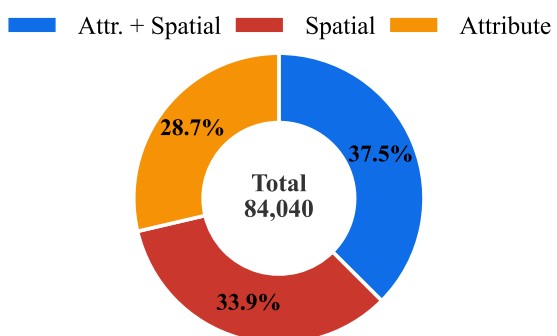

Figure 12. Ambiguity types distribution in 3D-Front.

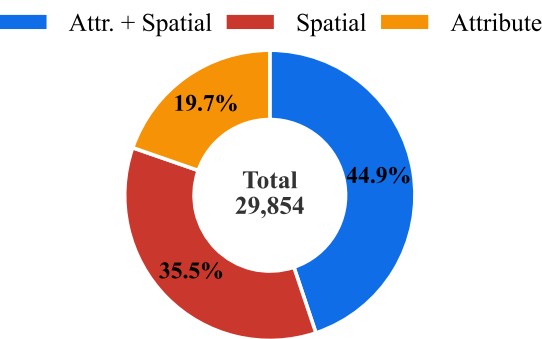

Figure 13. Ambiguity types distribution in ScanRefer.

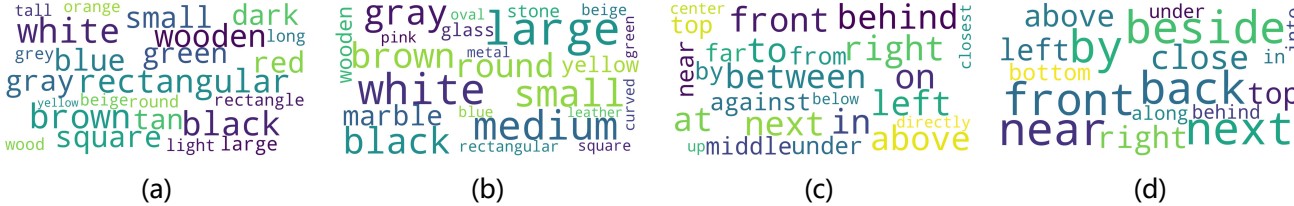

Figure 14. Missing words in ambiguous referring expressions across datasets and ambiguity types. **(a)** ScanRefer: frequently omitted visual attribute terms in Attribute queries. **(b)** 3D-FRONT: frequently omitted visual attribute terms in Attribute queries. **(c)** ScanRefer: frequently omitted spatial relation terms in Spatial Relation queries. **(d)** 3D-FRONT: frequently omitted spatial relation terms in Spatial Relation queries.

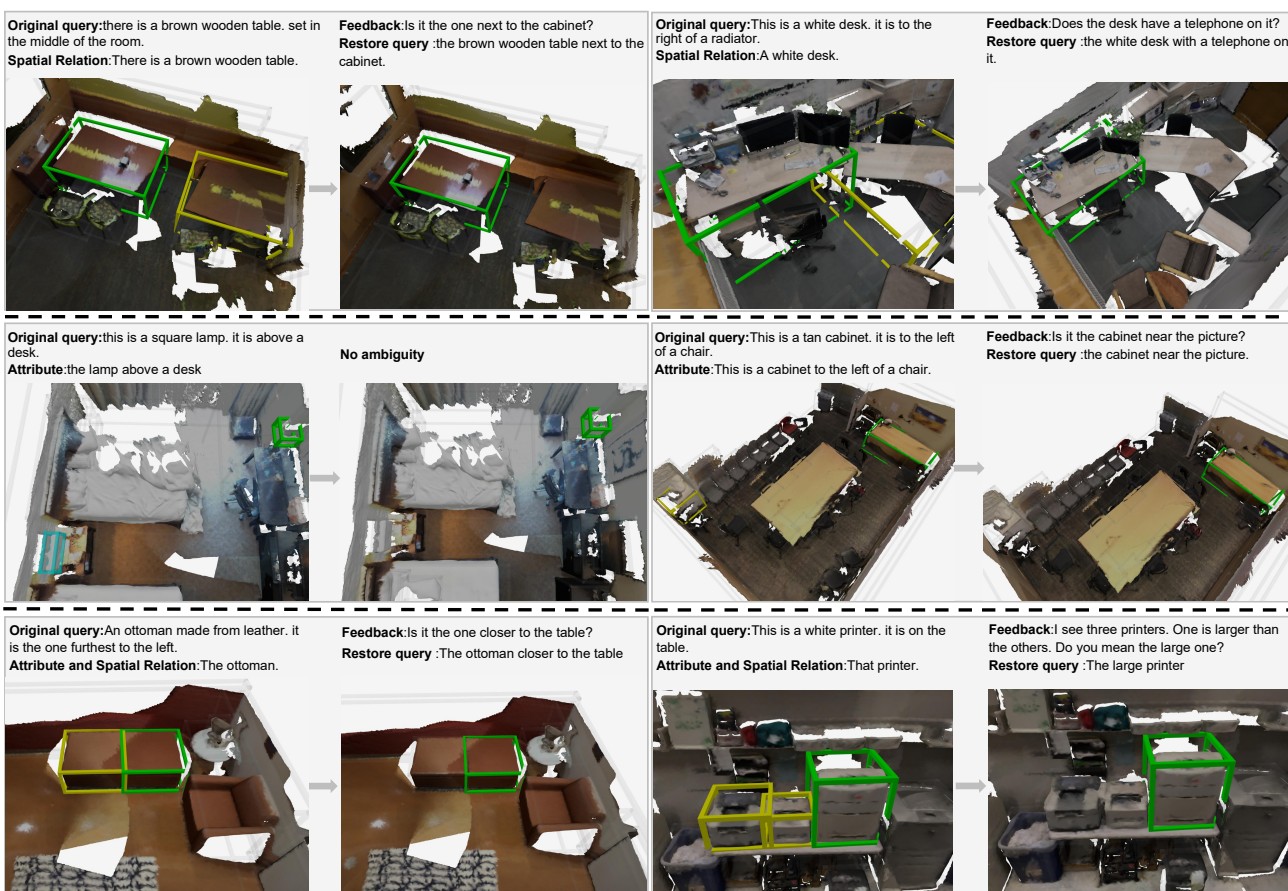

*Figure 15.* Examples of Real-world scenes from AmbiRefer3D. The green bounding box indicates the target object, the blue bounding box represents objects of the same type but do not meet the description, and the yellow bounding box represents objects of the same type that meet the description.

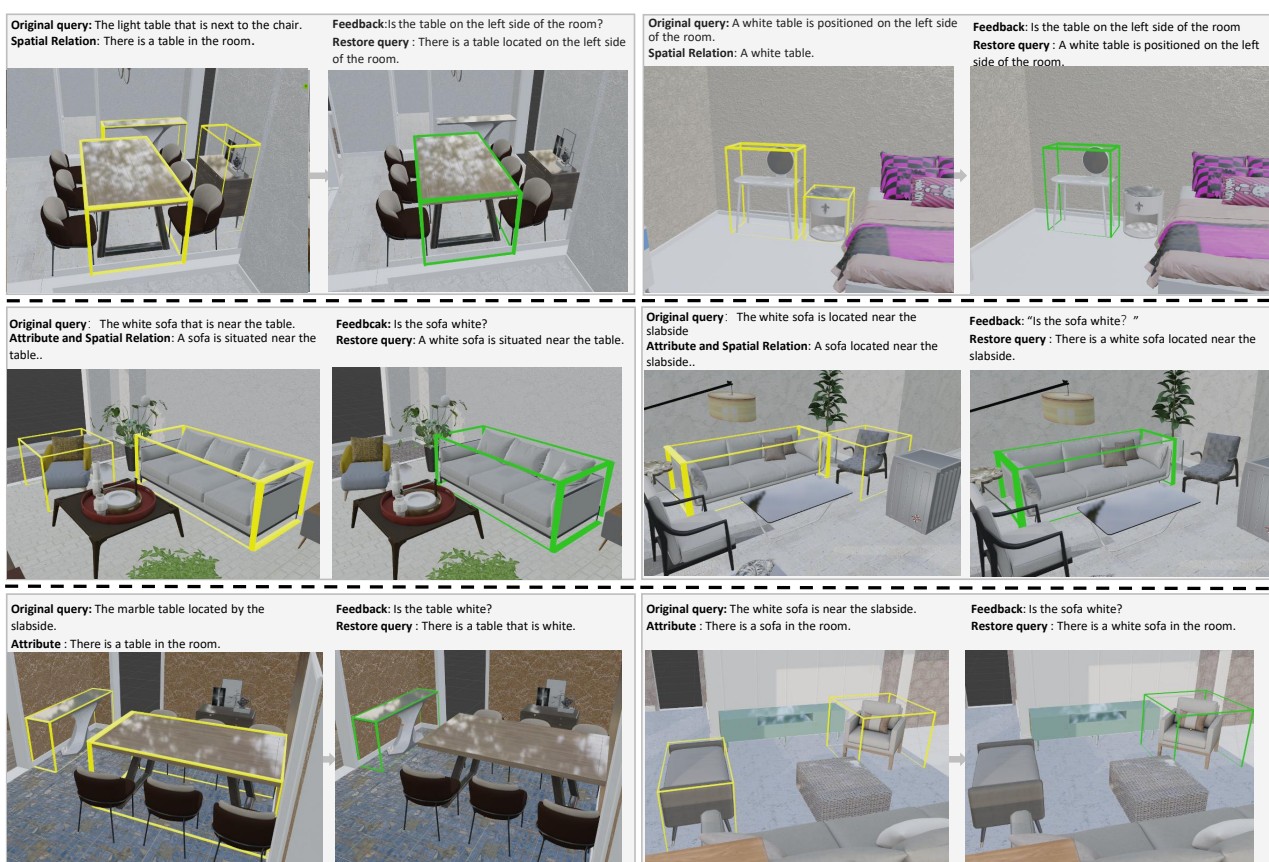

*Figure 16.* Examples of Synthetic scenes from AmbiRefer3D. The green bounding box indicates the target object, and the yellow bounding box represents objects of the same type that meet the description.

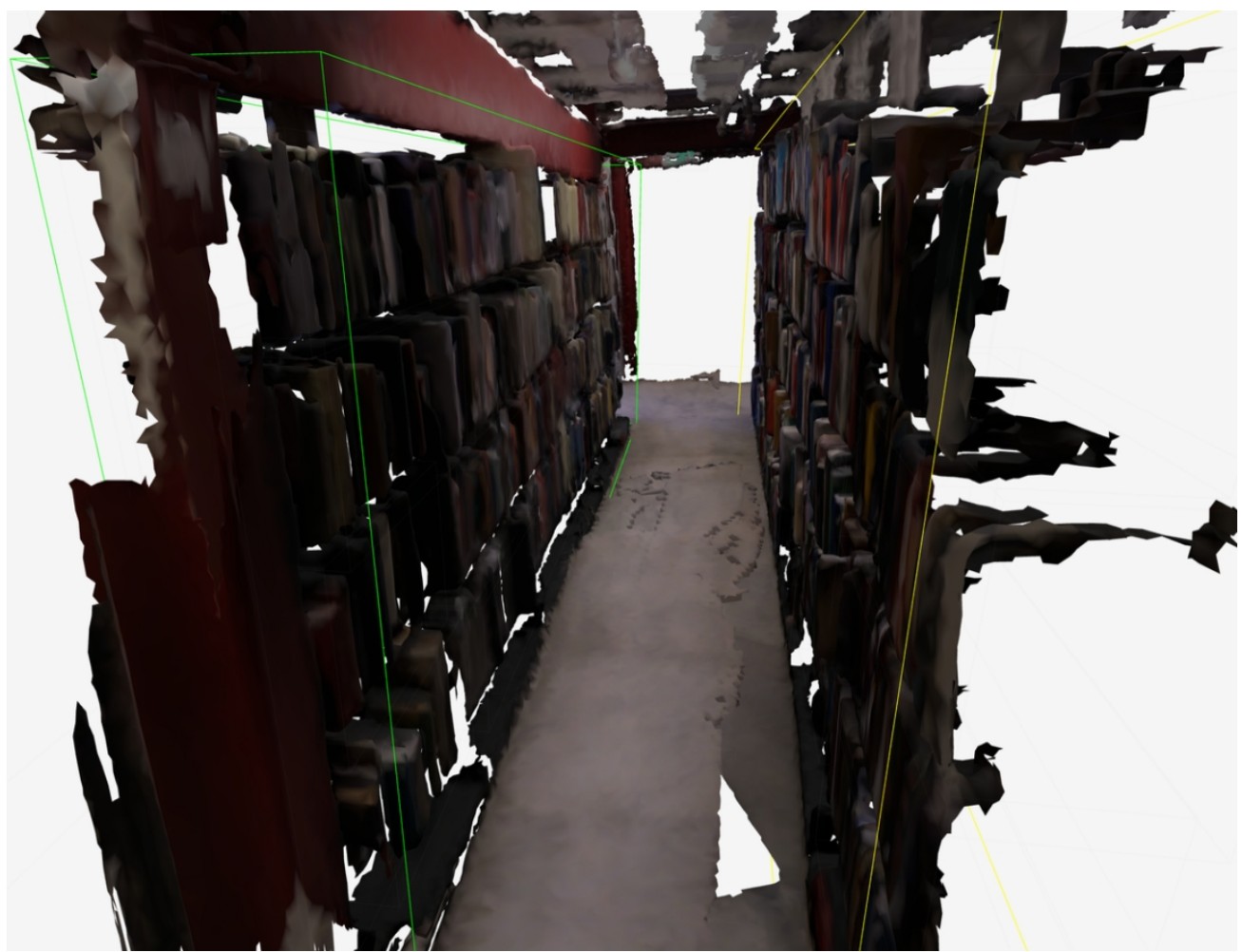

*Figure 17.* An example of a failed case in real-scenes.

