# OpenReview forum: "AmbiRefer3D: 3D Visual Grounding with Referential Ambiguity"
_ICML.cc/2026/Conference — ICML 2026 regular_

### Official Review · Reviewer_smKv · 2026-03-08

**Soundness:** 3
**Presentation:** 3
**Significance:** 3
**Originality:** 2
**Overall Recommendation:** 4
**Confidence:** 4

**Summary:**

This paper studies 3D visual grounding under referential ambiguity, where expressions may match multiple objects. A two-stage framework is proposed, combining candidate generation and interactive clarification-based disambiguation. A new dataset, AmbiRefer3D, is introduced, and experiments show improved performance in ambiguous scenarios.

**Compliance With Llm Reviewing Policy:**

Affirmed.

**Final Justification:**

All my concerns have been addressed. This paper is interesting and the author's responses are detailed.

**Key Questions For Authors:**

1.The constraint space is predefined. How scalable is this to open-ended attributes?

2.Can the model handle free-form human clarification responses?

3.Would performance degrade when clarification answers are noisy or partially incorrect?

4.Multi-view rendering + multiple clarification rounds introduce latency. What is the inference-time overhead compared to standard 3DVG?

**Limitations:**

From an ICML perspective, stronger theoretical analysis would significantly strengthen the contribution.

**Strengths And Weaknesses:**

Strengths

1. The paper addresses referential ambiguity in 3D visual grounding, a realistic yet underexplored setting. Reframing grounding as an interactive disambiguation process makes the task closer to real human–machine interaction.

2. The two-stage pipeline (candidate generation + interactive disambiguation) is conceptually clean. The decomposition into constraint selection, candidate filtering, and stopping prediction is intuitive and interpretable.

3. The AmbiRefer3D dataset is built through controlled semantic ablation and structured clarification generation, which provides a principled way to study ambiguity in 3D grounding.

Major Weaknesses

1.Limited technical novelty. While the task formulation is novel, the technical contributions are relatively limited.

2.Constrained clarification space. Clarifications are selected from a predefined discrete constraint space, which may limit scalability to open-vocabulary or free-form dialogue settings.

3.Limited task scope. The formulation assumes a single ground-truth target and does not consider none-target or multi-target scenarios.

4.Future research directions? For this type of ambiguity, it seems that interaction may be the only viable solution, since the speaker’s true intent cannot be directly known.

Minor Weaknesses

There are multiple instances where a single word appears alone on a line.

---

> ### Author Rebuttal · Authors · 2026-03-31
>
> **W3. Limited task scope**
>
> Our proposed framework can support both multi-target and none-target scenarios, albeit with specific structural adaptations.
>
> Multi-target grounding can be naturally supported by allowing multiple valid candidates and adapting the stopping criterion accordingly.
>
> For none-target scenarios, the current formulation does not explicitly handle cases where no object matches the query. Addressing this requires additional mechanisms, such as enabling the candidate filtering process to progressively reduce the candidate set to empty, or introducing explicit rejection criteria when no candidate satisfies the constraints. We consider this to be an important direction for future work.
>
> **W4. Future research directions?**
>
> We agree that interaction is essential under ambiguity, as the human’s intent cannot be fully inferred from a single expression.
> In our setting, ambiguity arises because the same expression may correspond to multiple valid objects in the scene, making it impossible to uniquely identify the target without additional information. Therefore, a clarifying process is necessary to resolve such ambiguity.
>
> In principle, this ambiguity could also be reduced by introducing strong priors, such as personalized preferences or habitual object usage. However, this would require access to user-specific or private contextual information, which is often unavailable or impractical to obtain.
>
> **Minor weakness**
>
> Thank you for pointing this out. We will carefully revise the formatting to remove such issues and improve overall readability in the final version.
>
> **Q1. How scalable to open-ended attributes?**
>
> Our framework can be naturally extended to open-ended attributes by leveraging stronger LLM/VLM modules or external knowledge sources to dynamically propose new constraints at inference time.
>
> **Q2. Can the model handle free-form responses?**
>
> Yes, our framework can certainly handle free-form human clarification responses. To process unconstrained answers, we can easily employ an LLM to extract key attributes or relationships from the free-form text and seamlessly map them into our candidate filtering module.
>
> However, we deliberately designed short, structured responses in our current formulation to minimize the user’s cognitive burden and reduce interaction complexity. Our method shifts the complexity to the model’s questioning side, where ambiguity is resolved through discriminative queries that progressively narrow the candidate space, thereby minimizing the burden on human responses.
>
> **Q3. Noisy clarification responses**
>
> Thank you for your valuable suggestion. Performance does degrade under noisy or incorrect clarification responses, as the interaction depends on feedback quality. To evaluate this, we inject controlled noise into answers:
>
> | **Answer Noise Rate** | **Acc** | **S@1** | **S@3** | **S@6** | **AvgR$\downarrow$** |
> |----------------------|--------|--------|--------|--------|--------|
> | 0%                   | **41.57** | **26.07** | **36.15** | **41.57** | **3.39** |
> | 20%                  | 37.42  | 21.09  | 31.84  | 37.42  | 3.67   |
> | 40%                  | 32.18  | 17.37  | 24.90  | 32.18  | 3.84   |
>
> Results show that increasing the answer noise rate from 0\% to 40\% leads to a gradual drop in Acc (41.57 → 32.18), confirming that noisy feedback negatively affects disambiguation.
> Despite this degradation, the increase in AvgR (3.39 → 3.84) suggests that the model can partially mitigate the impact of noise by engaging in additional rounds of interaction, allowing it to refine its predictions through iterative questioning. Developing additional mechanisms to handle noisy answers is an important direction for future work.
>
> **Q4. What is the inference-time overhead compared to standard 3DVG?**
>
> Thank you for valuable question. Our method introduces additional latency compared to one-shot 3DVG, mainly due to multi-view rendering and multi-round interaction. We provide a breakdown below:
>
> | **Method**         | **Views** | **AvgR** | **Rendering** | **Stage1** | **Stage2** | **ALL Latency** | **Relative Cost** |
> |------------------|----------|--------|-------------|-----------|-----------|--------------------|------------------|
> | M3DRef-CLIP      | -        | 1      | -           | -         | -         | 82 ms                | 1.00$\times$     |
> | M3DRef-CLIP+Ours | 3        | 3.42   | 3102 ms       | 82 ms       | 122 ms      | 3306 ms              | 40.31$\times$    |
>
> Importantly, the overhead is dominated by multi-view rendering, while the per-round computation remains lightweight. The interaction itself is therefore efficient, and the increased cost mainly comes from additional rounds rather than heavy model inference.
> Moreover, in practical settings where multi-view images are already available (e.g., robotics or embodied agents), the rendering cost can be largely eliminated, significantly reducing overall latency.

---

> > ### Author Rebuttal · Reviewer_smKv · 2026-04-03
> >
> > All my concerns have been addressed.

---

> > > ### Author Response · Authors · 2026-04-03
> > >
> > > We thank you again for your constructive comments, which have helped us improve this paper. We will incorporate the additional analyses and clarifications in the revised version.

---

### Official Review · Reviewer_89K4 · 2026-03-12

**Soundness:** 2
**Presentation:** 3
**Significance:** 2
**Originality:** 3
**Overall Recommendation:** 4
**Confidence:** 4

**Summary:**

This paper studies 3D visual grounding under referential ambiguity, where a language query may correspond to multiple objects in a scene. To address this, the authors formulate an interactive grounding task in which the model resolves ambiguity through multi-round clarification questions and answers until the target object can be uniquely identified.

To support the task, the paper introduces AmbiRefer3D, a dataset containing ambiguous referring expressions paired with clarification dialogues across indoor 3D scenes. The authors also propose an interactive grounding framework that first retrieves candidate objects and then iteratively selects clarification constraints and filters candidates based on the responses. Experiments show that integrating the proposed disambiguation module with existing grounding backbones improves grounding performance under ambiguous queries.

The main contributions are a new ambiguity-aware 3D grounding task, a large-scale dataset with clarification dialogues, and an interactive framework for multi-round disambiguation in 3D scenes.

**Compliance With Llm Reviewing Policy:**

Affirmed.

**Final Justification:**

While I still have some remaining concerns regarding the label mapping (e.g., the authors mention mapping nightstand in 3D-Front to the cabinet category, whereas nightstand is explicitly defined in NYU40), the rebuttal has addressed the majority of my questions. I have decided to raise my score to 4.

**Key Questions For Authors:**

1. Loss ablation: The method uses several loss terms. Could the authors provide ablation results showing the contribution of each component? This would clarify whether all newly introduced losses are necessary.

2. Semantic label alignment: AmbiRefer3D is built from datasets with different taxonomies (e.g., table / desk in ScanNet vs. table + desk in 3D-Front). Could the authors clarify how those label inconsistancy are addressed in the final constructed dataset?

3. Performance on unambiguous queries: Could the authors report the performance of the proposed framework on unambiguous referring expressions (e.g., ScanRefer)? This would help understand how robust the network on different scenarios.

**Limitations:**

The paper briefly discusses potential societal impacts and states that no significant negative impacts are expected. However, the discussion mainly focuses on general statements and does not thoroughly address several practical limitations of the proposed approach. For example, the framework assumes ideal user feedback during interaction, and the dataset construction relies heavily on automatically generated annotations whose reliability is not fully validated. The authors could strengthen this section by discussing these methodological limitations and the potential risks associated with automated dataset generation.

**Strengths And Weaknesses:**

Strengths:
1. The paper studies multi-turn object grounding in a conversational setting, which is a realistic extension of conventional single-shot 3D grounding. In practical applications (e.g., embodied agents or robot assistants), human instructions are often ambiguous and require clarification. The proposed task formulation therefore addresses an important limitation of existing benchmarks and models.
2. The proposed AmbiRefer3D dataset introduces ambiguity-aware referring expressions and clarification dialogues, which is a useful contribution for studying interactive grounding. The dataset is relatively large and covers multiple ambiguity types.
3. The proposed method is built on top of existing grounding backbones (e.g., M3DRef-CLIP and D-LISA), showing that the interactive disambiguation module can be integrated with prior methods. This makes the approach flexible and potentially reusable in future work.
4. The paper is easy to follow. The task formulation, dataset construction pipeline, and model architecture are clearly described.

Weakness:
1. The proposed model is trained using multiple losses, including grounding loss, constraint selection loss, candidate ranking loss, and stopping loss. However, the paper does not provide ablation studies isolating the contributions of these newly introduced components. It is therefore unclear whether all loss terms are necessary, which components contribute most to the final performance, and whether a simpler formulation could achieve similar results. A detailed ablation analysis would strengthen the technical validation.
2. Table 9 and Table 10 raise concerns about semantic label consistency across the datasets used to construct AmbiRefer3D. Different datasets use different semantic label definitions. For example, table and desk are defined separately in ScanNet but are merged as “table” in 3D-Front. In addition, some semantic labels in Table 9 seem to be defined at different levels of granularity, such as armchair belonging to chair, or bottle and box belonging to container. Labels such as seat, stool, and bench are also ambiguous and overlapping. These inconsistencies raise concerns about semantic alignment and dataset quality.
3. The dataset construction pipeline relies heavily on automated generation using vision-language models and large language models to produce attributes, referring expressions, and clarification dialogues. While the paper describes verification steps to reduce hallucinations, the reliability of the generated attributes and relations remains unclear. In addition, the pipeline relies on rendered object views from 3D meshes, which may be particularly noisy for real scanned scenes such as ScanNet where mesh quality is often bad. Without human verification or dataset quality analysis, it is difficult to assess the reliability of the generated annotations.
4. The paper does not evaluate the proposed framework on unambiguous referring expressions. While Appendix A shows that ambiguous queries degrade the performance of existing grounding models, it remains unclear how the proposed interactive framework behaves when the input query is already unambiguous (e.g., on ScanRefer). Evaluating this setting would help better understand the generalization capability of the proposed method.
5. The proposed framework may also be difficult to scale up due to its relatively complex architecture and multi-stage interaction design. In particular, the candidate filtering step permanently removes objects from the candidate set, meaning that objects filtered out in earlier rounds cannot be recovered later. If the true target object is mistakenly removed early in the interaction process, the model has no mechanism to recover it. This irreversible pruning strategy may make the system sensitive to early prediction errors and limit robustness in complex scenes with many objects.
6. Some parts of the presentation could be improved. Several minor typos appear in the text and tables (e.g., “conor” → “corner” in Fig. 6, redundant labels in Table 10).

---

> ### Author Rebuttal · Authors · 2026-03-31
>
> **W3. Dataset construction human verification**
>
> Thank you for your valuable suggestion. We have performed human verification results, which is detailed in Table 8 of Appendix D.3.
>
> **W5. Pruning strategy**
>
> Thank you for your suggestion. Yes, you are right, and permanently removing candidates could propagate early errors if the true target is mistakenly filtered out. However, in our method, this risk is mitigated in two key ways:
>
> **(1) Conservative filtering.** We adopt a relatively permissive threshold at each round to retain candidates with sufficiently high compatibility scores, prioritizing target preservation over aggressive pruning.
> **(2) Adaptive constraint selection.** The constraint selected at each round is optimized to be maximally discriminative given the current candidate set, which empirically reduces the likelihood of eliminating the true target in early stages.
>
> Although these strategies help mitigate target loss, they can not completely eliminate the risk of irreversible errors. A more promising direction would involve mechanisms to recover candidates that are previously filtered out, enabling the model to recall the true target when early mistakes occur. We believe that this is an important direction for future work.
>
> **W6. Typographical and formatting issues**
>
> Thank you for your thorough review! We will carefully proofread the paper, correct the noted typos to enhance readability further.
>
> **Q1. Loss ablation**
>
> Thank you for your valuable suggestion. We conduct a comprehensive ablation study on the loss components, and the results are shown as follows:
>
> |  $L_{sel}$ | $L_{cand}$ | $L_{stop}$ | **Acc** | **S@1** | **S@3** | **S@6** | **AvgR$\downarrow$** |
> |-----------|------------|------------|--------|--------|--------|--------|--------|
> |  ✓ | ✓ | ✗ | **42.36** | 25.89 | **36.44** | **42.36** | 3.74 |
> |  ✓ | ✗ | ✓ | 38.13 | **26.13** | 33.08 | 38.13 | 4.12 |
> | ✗ | ✓ | ✓ | 24.47 | 7.40 | 8.06 | 24.47 | 5.16 |
> | ✓ | ✓ | ✓ | 41.57 | 26.07 | 36.15 | 41.57 | **3.39** |
>
> The ablation results show that the three loss terms play complementary roles in the interactive disambiguation process.
>
> Removing $L_{stop}$ leads to the highest Acc, as the model is no longer constrained to stop early and can continue interaction until the target is correctly grounded. However, this comes at the cost of a higher AvgR, indicating reduced interaction efficiency. This reflects a trade-off between accuracy and efficiency, where $L_{stop}$ primarily regulates when to terminate the interaction.
>
> In contrast, removing either $L_{cand}$ or $L_{sel}$ results in clear performance degradation. Without $L_{cand}$, the model fails to effectively filter distractors, preventing progressive reduction of the candidate set. Without $L_{sel}$, the model cannot select informative constraints, leading to suboptimal and inefficient queries.
>
> These results highlight that effective interaction requires jointly optimizing constraint selection, candidate filtering, and stopping decisions.
>
>
> **Q2. Semantic label alignment**
>
> During dataset construction, we first map object categories from different source datasets into a unified semantic space following the common NYU40-aligned protocol, which is also consistent with the standard ScanNet/ScanRefer setting.
>
> For categories such as desk/table, we adopt a hierarchical alignment strategy. We first map objects into a unified coarse semantic space using their super-categories for cross-dataset consistency. We then examine subtype annotations, and if a subtype indicates a different coarse category from the initial super-category mapping, we further remap the object accordingly.
>
> **Q3. Performance on unambiguous referring expressions**
>
> Thank you for your valuable suggestion.Our proposed framework is specifically designed to tackle referential ambiguity. For standard unambiguous referring expressions (such as those in the ScanRefer dataset), the interactive disambiguation process is not required.
>
> In such cases, our method directly relies on the first-stage grounding module to produce the target prediction.
> Consequently, on unambiguous queries, our framework completely reduces to the base grounding model (e.g., M3DRef-CLIP or D-LISA), achieving the same performance as the original backbones without any degradation.

---

> > ### Author Rebuttal · Reviewer_89K4 · 2026-04-03
> >
> > Thank you for the additional clarifications. However, two issues remain unclear:
> >
> > **(Q2) Semantic label alignment.**
> >
> > The authors mention adopting NYU40 as a unified semantic space. However, to my understanding, only ScanNet officially provides mappings from raw labels to NYU40. It is unclear how this mapping is consistently extended to 3D-FRONT. Furthermore, Table 9 reflects NYU40-aligned categories from ScanNet, while Table 10 still shows mixed or finer-grained labels in 3D-FRONT, suggesting potential inconsistencies in the final taxonomy. Could the authors clarify in detail how labels from 3D-FRONT are mapped and reconciled with NYU40? At present, the concern regarding cross-dataset semantic consistency remains, which is a critical issue for the dataset contribution.
> >
> > **(Q3) Handling of unambiguous queries and system robustness.**
> >
> > the authors state that the framework reduces to the base grounding model when the input query is unambiguous. However, it is unclear whether this behavior is determined automatically by the model or requires external intervention (e.g., manually disabling the disambiguation module). In realistic deployment scenarios, user inputs may be either ambiguous or unambiguous, and the system should ideally make this determination autonomously. Requiring manual control would limit usability and generalization. Could the authors clarify whether the model includes a mechanism to detect ambiguity and adapt its behavior accordingly? This design choice is important for practical applicability and robustness.

---

> > > ### Author Response · Authors · 2026-04-04
> > >
> > > We sincerely thank you for these follow-up questions and hope that the additional explanation and experiments below further address your concerns.
> > >
> > > **Q2. Semantic label alignment**
> > >
> > > We would like to clarify that Table 10 presents the raw category statistics from the original 3D-FRONT annotations before alignment to provide a comprehensive overview of the source data’s diversity. These fine-grained or mixed labels do not reflect the final taxonomy.
> > >
> > > For 3D-FRONT, which provides both super-categories and category-level labels, we adopt a two-stage mapping strategy. We first perform a coarse mapping based on the super-category to align objects with the NYU40 semantic space. We then refine this mapping using the object-level category: if the category corresponds to a valid NYU40 class, we remap accordingly; otherwise, we retain the super-category mapping (e.g., objects such as *Children Cabinet*, *Nightstand*, and *Wardrobe* are mapped to *Cabinet*, while *Dining Table* and *Side Table* are mapped to \textit{Table}, ensuring semantically consistent alignment without collapsing into generic categories).
> > >
> > > This ensures that all instances from 3D-FRONT are projected into the same NYU40 semantic space used by ScanNet/ScanRefer. Categories absent in 3D-FRONT simply do not appear, rather than forming a different label system. This ensures semantic consistency across datasets. We hope that the above clarification addresses your concern and explicitly clarifies this distinction between raw data statistics and the final training taxonomy in the revised manuscript.
> > >
> > > **Q3. Handling of unambiguous queries and system robustness.**
> > >
> > > Thank for your detailed follow-up question. It should first be clarified if we know that the deployment environment contains only unambiguous queries, the interactive disambiguation process is not required, and the framework can reduce to the base grounding model and rely on the first-stage grounding module in this situation. However, in realistic deployment scenarios, user inputs may be either ambiguous or unambiguous, and their nature is often unknown beforehand. In such cases, our framework operates in a fully autonomous manner and does not require any manual intervention to distinguish between query types.
> > > In the Candidate Generation module, we use a confidence threshold to filter the initial candidate set. If the input query is sufficiently clear and results in only one candidate, the Stopping Head will naturally trigger an immediate termination.
> > > If more than one candidate remains and the model determines the current grounding state is not yet certain, further interaction is necessary to resolve potential referential ambiguity.
> > >
> > > To further evaluate system robustness and ability of handling unambiguous queries, we evaluated our framework on the ScanRefer dataset.
> > >
> > > | Method               | TS    | Acc           | S@1   | S@3   | S@6   | AvgR↓ |
> > > |----------------------|-------|---------------|-------|-------|-------|--------|
> > > | M3DRef-CLIP          | -     | 45.50         | -     | -     | -     |        |
> > > | M3DRef-CLIP+Ours     | 63.03 | 44.61 (-0.89%)| 36.10 | 40.47 | 44.61 | 2.46   |
> > > | D-LISA               |       | 46.30         | -     | -     | -     |        |
> > > | D-LISA+Ours          | 66.50 | 45.86 (-0.44%)| 35.74 | 42.15 | 45.86 | 1.92   |
> > >
> > > As shown in table, our framework achieves comparable performance to the base models on unambiguous queries, while introducing a small amount of additional interaction. The slight performance drop in early rounds mainly stems from the difference in prediction strategy. Instead of directly selecting the highest-scoring candidate as in the base models, our framework applies a relatively permissive threshold to determine the candidate set. As a result, even for unambiguous queries, multiple candidates may be retained in early stages.
> > > In addition, the interactive process introduces additional constraints that may be unnecessary for inherently unambiguous queries, which may impose extra burden on the decision process and slightly affect predictions. We expect this effect to diminish as the performance of the base grounding model improves, since stronger initial predictions make it less likely for irrelevant candidates to be retained.
> > >
> > > However, the model can still correctly resolve the target, demonstrating that our framework maintains reliable grounding ability on unambiguous queries. We argue that the slight delay caused by these additional rounds is practically acceptable in real-world deployment, as also discussed in response to Reviewer smKv (Q4).
> > > Overall, our framework not only improves performance under ambiguous queries, but also maintains reliable grounding ability on unambiguous queries without requiring manual intervention, supporting its robustness and practical applicability.

---

### Official Review · Reviewer_wGgM · 2026-03-13

**Soundness:** 2
**Presentation:** 3
**Significance:** 2
**Originality:** 3
**Overall Recommendation:** 4
**Confidence:** 4

**Summary:**

This paper introduces a new task: 3D visual grounding with iterative disambiguition from user, where the initial grounding expression may be ambiguous and matching multiple objects in the scene. To address this task, the authors propose an iterative object selection framework, that first selects candidate objects with a established 3D visual grounding model, and further filters candidates based on clearer prompts, until the unique target is selected. To train and evaluate this grounding framework, they also construct AmbiRefer3D dataset with three types of ambiguities with several useful metrics. Experiments show that the proposed iterative selection framework could achieve disambigution with iteratively incoming clearer grounding prompts.

**Compliance With Llm Reviewing Policy:**

Affirmed.

**Final Justification:**

The paper proposed a rather interesting problem setting and a corresponding method for iteratively disambiguation in 3D spatial reasoning scenario.
My initial concerns include the unclear inference pipepine, the need of comparison/combination with currently best performing 3D MLLMs, and the necessity of multi-round interactions.
The rebuttal have addressed most of my concerns in the response.
Regarding Q2, a more thorough integration of recent MLLM-based grounding methods as backbone within the proposed framework would further strengthen the contribution. This remains as a minor concern.
Overall, I'll consider raising my original score, and lean to weak accept.

**Key Questions For Authors:**

1. How exactly is $v_{target}$ (and the derived $\mu_{diff}$, and $\delta_{max}$) obtained during inference? If oracle target features are used at test time, the evaluation would be problematic. If an approximation is used (e.g., highest-scoring candidate, mean of candidates), please specify the details and provide ablations. This question is critical to my soundness assessmen, and a satisfactory answer would likely raise my score.

2. Could the authors integrate at least one recent MLLM-based 3D grounding method into the framework, or evaluate a simple MLLM-and-prompting baseline (e.g., providing multi-view images to a strong MLLM and asking it to disambiguate via conversation)? This would help contextualize the contribution against the current state of the art methods.

3. What is the performance of selecting all available constraints at once (single-round disambiguation) compared to the proposed multi-round approach? If single-round performs comparably, what is the justification for the added complexity of iterative interaction? More generally, could the authors provide ablation results on the constraint selection strategy (e.g., random selection, heuristic selection, all-at-once)?

**Limitations:**

It would be beneficial for the authors to further discuss the gap between the synthetic ambiguity construction and natural human language ambiguity.

**Strengths And Weaknesses:**

### Strengths

1. Well-motivated and practical problem. The observation that human instructions are often ambiguous and may require multi-round clarification is quite reasonable and not well studied in 3D grounding task.

2. Disambiguation process is interesting and interpretable. Unlike end-to-end LLM approaches recently, the structured pipeline (constraint selection -> iterative candidate filtering -> stopping) makes every step traceable. This step-by-step interpretability is interesting, and matters for practicality, where users may want to understand what the system is doing.

3. The proposed dataset, AmbiRefer3D, required significant engineering and efforts. It would be beneficial for the 3D vision-language learning research community and future work. The metrics (TS, S@K, Acc, etc.) seem to well capture the disambiguation efficiency and accuracy.

### Weaknesses

1. Probable information leak of $v_{target}$ in disambiguation. In Eq. (3), the constraint selector takes $v_{target}$, $\mu_{diff}$, and $\delta_{max}$ as inputs—all of which require knowing the ground-truth target object visual feature (as described in L179-L200). However, at inference time, the target object is exactly what we are trying to find. The paper does not explain how this is handled at test time, which makes it hard to assess the soundness of the reported experimental results.

2. Baselines are outdated and insufficient. The only two compared backbones in main experiments are M3DRef-CLIP (ICCV 2023) and D-LISA (NeurIPS 2024), which is not very recent works and this feels a bit out-dated. It would be beneficial to include more baseline comparisons: 1) using recent state-of-the-art LLM-based visual grounding models (e.g., Inst3DLMM [1], Video3D LLM [2]) with ambiguous **and/or** ground-truth queries, to understand how capable modern LLM-based methods already are at handling ambiguity natively; 2) using state-of-the-art visual grounding backbones combined with the proposed interactive disambiguation module, to demonstrate the claimed generality of the framework.

3. Unclear motivation for multi-round over single-round disambiguation. The paper does not convincingly justify why the constraint selection must be done in multiple rounds. Since at inference time, the user is the one providing answers/new contraints, there seems to be no practical difference between providing information across several rounds versus providing as much discriminative information as possible in a single iteration. A natural and strong baseline would be to simply select all relevant constraints at once and present them together—if this performs comparably, the multi-round design becomes unnecessary overhead while with the risk of error propagation. Moreover, the paper lacks ablation studies on the selection mechanism design (e.g., random selection, all-at-once selection, different selectors). In fact, the paper appears to lack substantive ablation studies altogether.

4. Minor: it would be more clear for readers to undertand the task, if the authors show what is a "constraint", and show certain dataset examples in Section 3 (Task definition). Typos (e.g., "march" at L313) also exist.

[1] Inst3D-LMM: Instance-Aware 3D Scene Understanding with Multi-modal Instruction Tuning, CVPR 2025.

[2] Video-3D LLM: Learning Position-Aware Video Representation for 3D Scene Understanding, CVPR 2025.

---

> ### Author Rebuttal · Authors · 2026-03-31
>
> **Q1. How $v_{target}$, $\mu_{diff}$ and $\delta_{max}$ obtained during inference?**
>
> **A1.** Thank you for aising this important question regarding the inference-time availability of $v_{target}$, as well as the derived statistics $\mu_{diff}$ and $\delta_{max}$.
> At inference time, the ground-truth target is not accessible. We approximate $v_{target}^{(r)}$ using the highest-scoring candidate:
> $\hat{o}^{(r)}= \mathop{\arg\max}\limits_{o_i \in C^{(r)}}$ score$^{(r)}(o_i), \quad v_{target}^{(r)}=v_{\hat{o}}^{(r)}$.
> When $r=1$, score$^{(r)}$ are obtained from the candidate generation, where each score represents the confidence of a candidate.
> For $r>1$, after the selector predicts the clarification constraint $c^{(r-1)}$, the mask head produces a compatibility score $m_i^{(r-1)}$ for each $o_i \in C^{(r-1)}$. We use $m_i^{(r-1)}$ as a round-wise confidence.
> The statistics $\mu_{diff}$ and $\delta_{max}$ are then computed with respect to $v_{target}^{(r)}$.
>
> To evaluate the impact of different approximations, we conduct ablation studies with alternative selection strategies for approximating
> $v_{target}$, including:(1) Random, (2) Mean Aggregation, and (3) Highest-scoring(Ours).
>
> | **Selection of $v_{target}$** | **Acc** | **S@1** | **S@3** | **S@6** | **AvgR$\downarrow$** |
> |--------------------------|--------|--------|--------|--------|--------|
> | Random         | 28.92  | 10.41  | 16.88  | 28.92  | 4.16   |
> | Mean Aggregation       | 21.09  | 5.71   | 11.14  | 21.09  | 4.34   |
> | Highest-scoring(**Ours**)| **41.57** | **26.07** | **36.15** | **41.57** | **3.39** |
>
> Results show that the highest-scoring selection achieves the best performance. This confirms that our strategy effectively identifies the most informative target representation, achieving better grounding performance.
>
> **Q2. MLLM-and-prompting baseline**
>
> **A2.** Thank you for your insightful suggestion.
> We can integrate MLLM-based 3D grounding methods as the candidate generation in our framework.
> Due to the limited time available during the rebuttal period, we do not train such integrated models. Instead, we build two MLLM-and-prompting baselines using Inst3D-LMM and Video-3D LLM. At each round, the MLLM asks a clarifying question, which is used to query Qwen3-VL-32B to answer strictly based on the target object. This repeates until the MLLM determines that the query is not ambiguous or a maximum of rounds is reached, after which MLLM outputs the final bounding box.
> We evaluate under the Mixture ambiguity setting, and the results are shown as follows:
>
> | **Method** | **Acc** | **S@1** | **S@3** | **S@6** | **AvgR$\downarrow$** |
> |----------------|--------|--------|--------|--------|--------|
> | Inst3D-LMM     | 26.80  | 10.55  | 18.70  | 26.80  | 4.85   |
> | Video-3D LLM   | 22.35  | 8.10   | 15.42  | 22.35  | 5.24   |
> | **Ours**       | **41.57** | **26.07** | **36.15** | **41.57** | **3.39** |
>
> Our method achieves the best performance across all metrics. This demonstrates the effectiveness of our framework for resolving ambiguous 3DVG.
>
> **Q3. Constraint selector strategy**
>
> **A3.** Selecting all available constraints at once is generally ineffective, as all constraints may be mutually incompatible or introduce conflicting signals when applied jointly. However, selecting multiple constraints simultaneously is a reasonable stragety to explore. To further validate this and systematically study the impact of different selection strategies, we compare four settings: (1) Random Selection, (2) Heuristic Selection (greedy reduction candidates), (3) Top-$K$ Selection ($K=3$ constraints applied in a single round), and (4) Ours.
>
> | **Constraint Selector Strategy** | **Acc** | **S@1** | **S@3** | **S@6** | **AvgR$\downarrow$** |
> |---------------------------------|--------|--------|--------|--------|--------|
> | Random Selection                | 23.31  | 8.53   | 9.16   | 23.31  | 4.88   |
> | Heuristic Selection             | 33.76  | 24.48  | 30.21  | 33.76  | 2.76   |
> | All-at-once                     | 37.88  | **30.42** | 35.69  | 37.88  | **1.89** |
> | **Ours**                        | **41.57** | 26.07  | **36.15** | **41.57** | 3.39   |
>
> We observe that Top-$K$ selection can accelerate early-stage disambiguation, achieving relatively strong performance at S@1 (30.42).
> However, its gains at later rounds remain limited.
> This is because our dataset is inherently designed with multiple complementary constraints,
> where each instance typically involves three or more constraints that must be jointly satisfied.
> When top-ranked constraints are applied together in one round, the model can benefit from stronger initial pruning without accumulated interaction error. However, as more constraints are jointly introduced, their effects may potentially interfere with each other, limiting further improvement at S@3 and S@6.
>
> **Minor weakness**
>
> Thank for the suggestion. We will add corresponding information in Section 3 and fix the noted typos in the revision.

---

> > ### Author Rebuttal · Reviewer_wGgM · 2026-04-04
> >
> > I appreciate the authors for providing further clarifications and empirical results, and they have addressed most of my concerns.
> > Regarding Q2, a more thorough integration of recent MLLM-based grounding methods as backbone within the proposed framework would further strengthen the contribution.
> > This remains as a minor concern.
> > Overall, I'll consider raising my score.

---

> > > ### Author Response · Authors · 2026-04-04
> > >
> > > We sincerely appreciate your positive feedback and your decision to raise your score. We are truly grateful for your thoughtful comments and constructive suggestions throughout the review process. We will carefully revise the paper accordingly and further strengthen the final version based on your valuable feedback.

---

### Decision · Program_Chairs · 2026-04-30

**Decision:**

Accept (regular)

**Comment:**

This paper introduces a timely and practically relevant setting for 3D visual grounding by explicitly modeling referential ambiguity and resolving it through interactive clarification. Reviewers found the problem formulation well motivated, and viewed the dataset and benchmark contribution as the main strengths of the work. The proposed framework is also clear and reasonably convincing, though several reviewers initially raised concerns about inference-time soundness, limited baselines, insufficient ablations, dataset label consistency, and efficiency. The authors addressed most of these concerns in the rebuttal with additional experiments and clearer explanations, which improved confidence in the work. Although the method itself is not highly novel in isolation and some details could still be strengthened in the final version, the paper opens up an interesting direction and provides useful resources for future work. I therefore recommend acceptance.